# Retrieval Augmented Time Series Forecasting

**Sungwon Han** [* 1 2]  **Seungeon Lee** [* 2]  **Meeyoung Cha** [3]  **Sercan Ö. Arik** [4]  **Jinsung Yoon** [4]

## Abstract

Time series forecasting uses historical data to predict future trends, leveraging the relationships between past observations and available features. In this paper, we propose RAFT, a retrieval-augmented time series forecasting method to provide sufficient inductive biases and complement the model's learning capacity. When forecasting the subsequent time frames, we directly retrieve historical data candidates from the training dataset with patterns most similar to the input, and utilize the future values of these candidates alongside the inputs to obtain predictions. This simple approach augments the model's capacity by externally providing information about past patterns via retrieval modules. Our empirical evaluations on ten benchmark datasets show that RAFT consistently outperforms contemporary baselines with an average win ratio of 86%.

## 1. Introduction

Accurately predicting future trends is crucial for making informed decisions in various fields. Time series forecasting, which analyzes past data to anticipate future outcomes, plays a vital role in diverse areas like climate modeling (Zhu & Shasha, 2002), energy (Martín et al., 2010), economics (Granger & Newbold, 2014), traffic flow (Chen et al., 2001), and user behavior (Benevenuto et al., 2009). By providing reliable predictions, it empowers us to develop effective strategies and policies across these domains.

Over the past decade, deep learning models such as CNNs (Bai et al., 2018; Borovykh et al., 2017) and RNNs (Hewamalage et al., 2021) have proven their

---

[*]Equal contribution [1]Department of AI Convergence, GIST, Gwangju, South Korea. This work is done while the author was in KAIST. [2]School of Computing, KAIST, Daejeon, South Korea [3]Data Science for Humanity Group, Max Planck Institute for Security and Privacy, Bochum, Germany [4]Google Cloud AI, Sunnyvale, United States. Correspondence to: Jinsung Yoon <jinsungyoon@google.com>.

*Proceedings of the 42nd International Conference on Machine Learning*, Vancouver, Canada. PMLR 267, 2025. Copyright 2025 by the author(s).

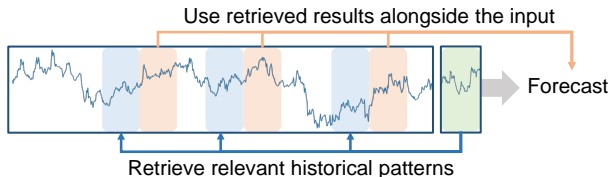

*Figure 1.* Illustration of a motivating example of retrieval in time-series forecasting.

effectiveness in capturing patterns of change in historical observations, leading to the development of various deep learning models tailored for time series forecasting. Especially, the advent of attention-based transformers (Vaswani et al., 2017) has made a significant impact on the time series domain. The architecture has shown to be effective in modeling dependencies between inputs, resulting in variants like Informer (Zhou et al., 2021), AutoFormer (Wu et al., 2021), and FedFormer (Zhou et al., 2022). Additionally, recent methods utilize time series decomposition (Wang et al., 2023), which isolates trends or seasonal patterns, and multi-periodicity analysis which involves downsampling/upsampling of the series at various periods (Lin et al., 2024; Wang et al., 2024). Furthermore, lightweight models like multi-layer perceptrons (MLP) have demonstrated strong performance along with these decomposition techniques and multi-periodicity analysis (Chen et al., 2023; Zeng et al., 2023; Zhang et al., 2022).

However, real-world time series exhibit complex, non-stationary patterns with varying periods and shapes. These patterns may lack inherent temporal correlation and arise from non-deterministic processes, resulting in infrequent repetitions and diverse distributions (Kim et al., 2021). This raises concerns about the effectiveness of models in extrapolating from such infrequent patterns. Moreover, the advantages of indiscriminately memorizing all patterns, including noisy and uncorrelated ones, are questionable in terms of both generalizability and efficiency (Weigend et al., 1995).

We show an advancement in time-series forecasting models by expanding the models' capacity (implicitly via the trained weights) to learn patterns. We directly provide information about historical patterns that are complex to learn, as a way of bringing relevant information via the input to reduce the burden on the forecasting model. Inspired

by the retrieval-augmented generation (RAG) approaches used in large language models (Lewis et al., 2020), our method retrieves similar historical patterns from the training dataset based on given inputs and utilizes them along with the model's learned knowledge to forecast the next time frame (see Figure 1).

Our new approach, Retrieval-Augmented Forecasting of Time-series (RAFT), offers two key advantages. First, by directly utilizing retrieved information, the useful patterns from the past become explicitly available at inference time, rather than utilizing them via the learned information in model weights. Learning hence covers patterns that lack temporal correlation or do not share common characteristics with other patterns, thereby reducing the learning burden and enhancing generalizability. Second, even if a pattern rarely appears in historical data and is difficult for the model to memorize, the retrieval module allows the model to easily leverage historical patterns when they reappear (Miller et al., 2024; Laptev et al., 2017).

We demonstrate that the proposed judiciously-designed inductive bias, implemented through a simple retrieval module, enables an MLP architecture to achieve strong forecasting performance. Inspired by existing literature that downsamples series at various period intervals (Lin et al., 2024; Wang et al., 2024), RAFT also generates multiple series by downsampling the given series at different periods and attaches a retrieval module to each series. This allows for effectively capturing both short-term and long-term patterns for more accurate forecasting. As demonstrated on ten time-series benchmark datasets, RAFT outperforms other contemporary baselines with an average win ratio of 86%. Overall, our contributions can be summarized as follows:[1]

- We propose a retrieval-augmented time series forecasting method, RAFT, which retrieves observations with similar temporal patterns from the training dataset and effectively leverage retrieved patterns for future predictions.

- Our empirical studies on ten different benchmark datasets show that RAFT outperforms other contemporary baselines with an average win ratio of 86%.

- We further explore the scenarios where retrieval modules can be beneficial for forecasting by conducting analyses using synthetic and real-world datasets.

## 2. Related Work

### 2.1. Deep learning for time-series forecasting

A large body of research employs deep learning for time-series forecasting. Existing methods can be broadly

categorized based on the employed architecture. Prior to the advent of transformers (Vaswani et al., 2017), time series analysis often relied on CNNs to capture local temporal patterns (Bai et al., 2018; Borovykh et al., 2017) or RNNs to model sequential dependencies (Hewamalage et al., 2021). Following the advent of transformers, several approaches emerged to better tailor the transformer architecture for time-series forecasting. For example, LogTrans (Li et al., 2019) used a convolutional self-attention layer, while Informer (Zhou et al., 2021) employed a ProbSparse attention module along with a distilling technique to efficiently reduce network size. Both Autoformer (Wu et al., 2021) and FedFormer (Zhou et al., 2022) decomposed time series into components like trend and seasonal patterns for prediction.

Despite advancements in transformer-based models, (Zeng et al., 2023) reported that even a simple linear model can achieve strong forecasting performance. Subsequently, lightweight MLP-based time-series models such as TiDE (?), TSMixer (Chen et al., 2023), and TimeMixer (Wang et al., 2024) were introduced with the advantages in both forecasting latency and training efficiency. These models utilize various approaches such as series decomposition similar to transformer-based studies (Zeng et al., 2023) or introduced multi-periodicity analysis by downsampling or upsampling the series at various period intervals (Lin et al., 2024), to accurately extract the relevant information from time-series for MLPs to effectively fit on them. Recently, several studies have constructed a large time-series databases to build large foundation models, achieving strong zero-shot and few-shot performance (Das et al., 2024; Woo et al., 2024).

Our proposed RAFT is based on a shallow MLP architecture, following simplicity and efficiency motivations. Through the retrieval module, the model retrieves subsequent patterns that follow the patterns most similar to the current input from the single time series, allowing it to reference past patterns for future predictions without the burden of memorizing all temporal patterns during training. Our retrieval differs from transformer variants that typically learn relationships only within a fixed lookback window. RAFT goes beyond the lookback window by retrieving relevant data points from the entire time series and incorporating them into the input.

### 2.2. Retrieval augmented models

Retrieval-augmented models typically work by first retrieving relevant instances from a dataset based on a given input. Then, they combine the input with these retrieved instances to generate a prediction. Retrieval-augmented generation (RAG) in natural language domain is an active research area that utilizes this scheme. (Lewis et al., 2020; Guu et al., 2020). RAG retrieves document chunks from external

---

[1] Code is in https://github.com/archon159/RAFT

corpora that are relevant to the input task, helping large language models (LLMs) generate responses related to the task without hallucination (Shuster et al., 2021; Borgeaud et al., 2022). This not only supplements the LLM's limited prior knowledge but also enables the LLM to handle complex, knowledge-intensive tasks more effectively by providing additional information from the retrieved documents (Gao et al., 2023).

Beyond natural language processing, retrieval-augmented models have also been used to solve structured data problems. A simple illustrative example is the K-nearest neighbor model (Zhang, 2016). Other approaches have introduced kernel-based neighbor methods (Nader et al., 2022), prototype-based approaches (Arik & Pfister, 2020), or considered all training samples as retrieved instances (Kossen et al., 2021). More recently, models leveraging attention-like mechanisms have incorporated the similarity between retrieved instances and the input into the prediction, achieving superior performance compared to traditional deep tabular models (Gorishniy et al., 2024). There also exists a method that has explored the potential of retrieving similar entities in time-series forecasting, involving multiple time series entities (Iwata & Kumagai, 2020; Yang et al., 2022). Assuming the training set contains various types of time series entities, they aggregate the information needed for each entity's prediction based on the similarities across all time series entities.

In this paper, we aim to demonstrate that retrieval can be effective, even when applied to the single time-series. Similar to how RAG supplements LLMs with additional information for knowledge-intensive tasks, our approach seeks to reduce the learning complexity in time-series forecasting. Instead of forcing the model to learn every possible complex pattern, the retrieval module provides information that simplifies the learning process.

## 3. Method

### 3.1. Overview

**Problem formulation.** Given a single time series $\mathbf{S} \in \mathbb{R}^{C \times T}$ of length $T$ with $C$ observed variates (i.e., channels), RAFT utilizes historical observation $\mathbf{x} \in \mathbb{R}^{C \times L}$ to predict future values $\mathbf{y} \in \mathbb{R}^{C \times F}$ that is close to the actual future values $\mathbf{y}_0 \in \mathbb{R}^{C \times F}$. $L$ denotes look-back window size and $F$ denotes forecasting window size.

Given an input $\mathbf{x}$, RAFT utilizes a retrieval module to find the most relevant patch from $\mathbf{S}$. Then, the subsequent patches of the relevant patch are retrieved as additional information for forecasting. The retrieval process follows an attention-like structure, where the importance weights are calculated based on the similarity between the input and the patches, and the retrieved patches are aggregated through a weighted sum (Sec. 3.2). The main difference of our model from attention-based forecasting models, such as transformers, lies in its ability to retrieve relevant data from the entire time series rather than relying on a fixed lookback window. Since the time series shows distinct characteristics across periods, we utilize the retrieval modules into multiple periods. RAFT generates multiple time series by downsampling the time series $\mathbf{S}$ with different periods and applies the retrieval module to each time series. The retrieval results from multiple series are processed through linear projection and aggregated by summation. Finally, the input and the aggregated retrieval result are concatenated and passed through a linear model to produce the final prediction (Sec. 3.3). Details of each component are described below.

### 3.2. Retrieval module architecture

We transform the time series $\mathbf{S}$ to be appropriate for retrieval. First, we find all *key* patches within $\mathbf{S}$ that are to be compared with given $\mathbf{x} \in \mathbb{R}^{C \times L}$. Using the sliding window method of stride $1$[2], we extract patches of window size $L$ and define this collection as $\mathcal{K} = \{\mathbf{k}_1, ..., \mathbf{k}_{T-(L+F)+1}\}$, where $i$ indicates the starting time step of the patch $\mathbf{k}_i \in \mathbb{R}^{C \times L}$. Note that any patch that overlaps with the given $x$ must be excluded from $\mathcal{K}$ during the training phase. Then, we find all *value* patches that sequentially follow each key patch $\mathbf{k}_i \in \mathcal{K}$ in the time series. We define the collection of value patches as $\mathcal{V} \in \{\mathbf{v}_1, ..., \mathbf{v}_{T-(L+F)+1}\}$, where each $\mathbf{v}_i \in \mathbb{R}^{C \times F}$ sequentially follows after $\mathbf{k}_i$ in the time series.

After preparing the key patch set $\mathcal{K}$ and value patch set $\mathcal{V}$ for retrieval, we use the input $\mathbf{x}$ as a *query* to retrieve similar key patches along with their corresponding value patches with following steps. We first account for the distributional deviation between the query, key, and value patches used in the retrieval process. Let us define $\mathbf{x} = \{\mathbf{x}^t\}_{t \in \{1, ..., L\}}$, where $\mathbf{x}^t \in \mathbb{R}^C$ denotes the values of $C$ variates at $t$-th time step within the input $\mathbf{x}$ (i.e., $\mathbf{x}^t = \{x_1^t, ..., x_C^t\}$). Inspired by existing literature (Zeng et al., 2023), we treat the final time step value in each patch as an offset and subtract this value from the patch as a form of preprocessing to make the patterns more meaningful to compare:

$$\hat{\mathbf{x}} = \{\mathbf{x}^t - \mathbf{x}^L\}_{t \in \{1, ..., L\}}, \tag{1}$$

where $\hat{\mathbf{x}}$ represent the input queries with the offset subtracted. Similarly, we subtract the offset from all key patches $\mathbf{k}_i \in \mathcal{K}$ and $\mathbf{v}_i \in \mathcal{V}$, denoting them as $\hat{\mathbf{k}}_i \in \hat{\mathcal{K}}$ and $\hat{\mathbf{v}}_i \in \hat{\mathcal{V}}$, respectively. Then, we calculate the similarity $\rho_i$ between given $\hat{\mathbf{x}}$ and all key patches in $\hat{\mathcal{K}}$ using similarity function $s$:

$$\rho_i = s(\hat{\mathbf{x}}, \hat{\mathbf{k}}_i), \quad \hat{\mathbf{k}}_i \in \hat{\mathcal{K}}. \tag{2}$$

Here, we use Pearson's correlation as a similarity function $s$ to exclude the effects of scale variations and value offsets

---

[2] The stride can be adjusted according to the demand of computational efficiency.

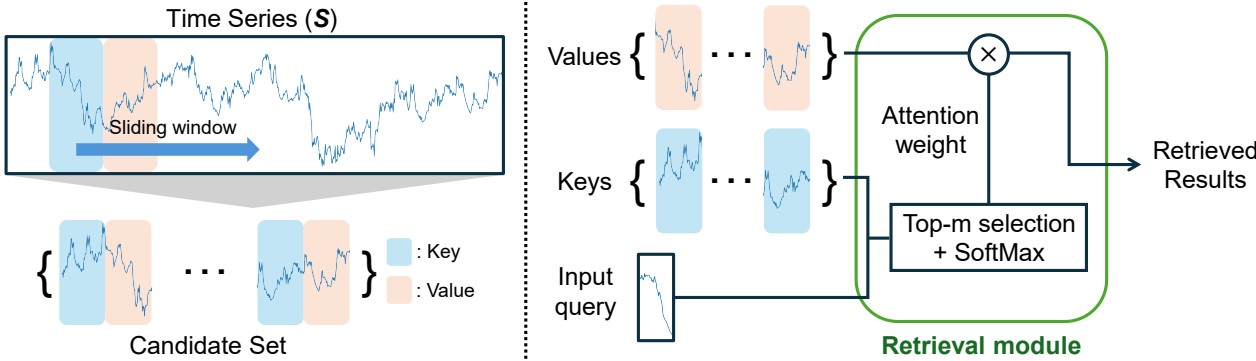

*Figure 2.* Illustration of retrieval module architecture. First, we consider consecutive time frames from the entire time series **S** as key-value pairs and construct a candidate set using a sliding window approach. Given an input time series as the query, the retrieval module computes the similarity between the query and the keys in the candidate set that do not overlap temporally. Based on the similarity, the top-$m$ candidates are selected, and attention weights are calculated via SoftMax. The final result is obtained through a weighted sum of the corresponding values.

in the time series, focusing on capturing the increasing and decreasing tendencies[3]. We then retrieve the patches with top-$m$ correlation values:

$$\mathcal{J} = \arg \text{top-}m \left( \{\rho_i \mid 1 \le i \le |\hat{\mathcal{K}}|\} \right), \quad (3)$$

where $\mathcal{J}$ denotes the indices of top-$m$ patches. Given temperature $\tau$, we calculate the weight of value patches with following equation:

$$w_i = \begin{cases} \frac{\exp(\rho_i/\tau)}{\sum_{j \in J} \exp(\rho_j/\tau)}, & \text{if } i \in \mathcal{J} \\ 0. & \text{otherwise} \end{cases} \quad (4)$$

Note that this is equivalent to conduct SoftMax only with top-$m$ correlation values. Finally, we obtain the final retrieval result $\tilde{\mathbf{v}} \in \mathbb{R}^{C \times F}$ as the weighted sum of value patches:

$$\tilde{\mathbf{v}} = \sum_{i \in \{1, \dots, |\hat{\mathcal{V}}|\}} w_i \cdot \hat{\mathbf{v}}_i. \quad (5)$$

Figure 2 illustrates the architecture of our retrieval module.

### 3.3. Forecast with retrieval module

**Single period.** Consider the given input $\mathbf{x} \in \mathbb{R}^{C \times L}$ and the retrieved patch $\tilde{\mathbf{v}} \in \mathbb{R}^{C \times F}$. Similar to the retrieval module, we subtract the offset from $\mathbf{x}$ and define $\hat{\mathbf{x}}$ as the input with the offset removed. Next, we concatenate $f(\hat{\mathbf{x}})$ with $g(\tilde{\mathbf{v}})$, and process concatenated result through $h$ to obtain $\hat{\mathbf{y}}$:

$$\hat{\mathbf{y}} = h(f(\hat{\mathbf{x}}) \oplus g(\tilde{\mathbf{v}})), \quad (6)$$

---

[3]See Appendix C.1 for comparison results with different similarity metrics.

where linear projections $f$ maps $\mathbb{R}^L$ to $\mathbb{R}^F$, $g$ maps $\mathbb{R}^F$ to $\mathbb{R}^F$, $h$ maps $\mathbb{R}^{2F}$ to $\mathbb{R}^F$, and $\oplus$ represents concatenation operation.

**Multiple periods.** Time series at different periods display unique characteristics – patterns in a small time window typically reveal local patterns, while patterns in a large time window might correspond to global trends. We extend the retrieval process to consider $n$ periods $\mathcal{P}$. For each $p \in \mathcal{P}$, we downsample the query $\mathbf{x}$, all key patches in $\mathcal{K}$, and all value patches in $\mathcal{V}$ of period 1 by average pooling with period $p$. This results in $\mathbf{x}^{(p)} \in \mathbb{R}^{C \times \lfloor \frac{L}{p} \rfloor}$, $\mathcal{K}^{(p)}$, and $\mathcal{V}^{(p)}$ as the respective query, key patch set, and value patch set for period $p$, where a key patch $\mathbf{k}_i^{(p)} \in \mathbb{R}^{C \times \lfloor \frac{L}{p} \rfloor}$ and a value patch $\mathbf{v}_i^{(p)} \in \mathbb{R}^{C \times \lfloor \frac{F}{p} \rfloor}$. Then, we conduct the retrieval process described in Sec. 3.2 using $\mathbf{x}^{(p)}$, $\mathcal{K}^{(p)}$, and $\mathcal{V}^{(p)}$, and obtain the retrieval result $\tilde{\mathbf{v}}^{(p)} \in \mathbb{R}^{C \times \lfloor \frac{F}{p} \rfloor}$ for each $p$. Each $\tilde{\mathbf{v}}^{(p)}$ is processed through a linear layer $g^{(p)}$ to project all retrieval results in the same embedding space, mapping $\mathbb{R}^{\lfloor \frac{F}{p} \rfloor}$ to $\mathbb{R}^F$, respectively. Finally, we concatenate $\hat{\mathbf{x}}$ with sum of linear projections and process it through linear predictor $h$, which replaces Eq. 6 to following equation:

$$\hat{\mathbf{y}} = h(f(\hat{\mathbf{x}}) \oplus \sum_{p \in \mathcal{P}} g^{(p)}(\tilde{\mathbf{v}}^{(p)})) \quad (7)$$

Denoting $\hat{\mathbf{y}}^t$ as the value at the $t$-th time step within $\hat{\mathbf{y}}$, we restore the original offset by adding $\mathbf{x}^L$ to $\hat{\mathbf{y}}$, resulting in the final forecast $\mathbf{y}$:

$$\mathbf{y} = \{\hat{\mathbf{y}}^t + \mathbf{x}^L\}_{t \in \{1, \dots, F\}}. \quad (8)$$

We train the model by minimizing the following MSE loss:

$$\mathcal{L} = \text{MSE}(\mathbf{y}, \mathbf{y}_0) \quad (9)$$

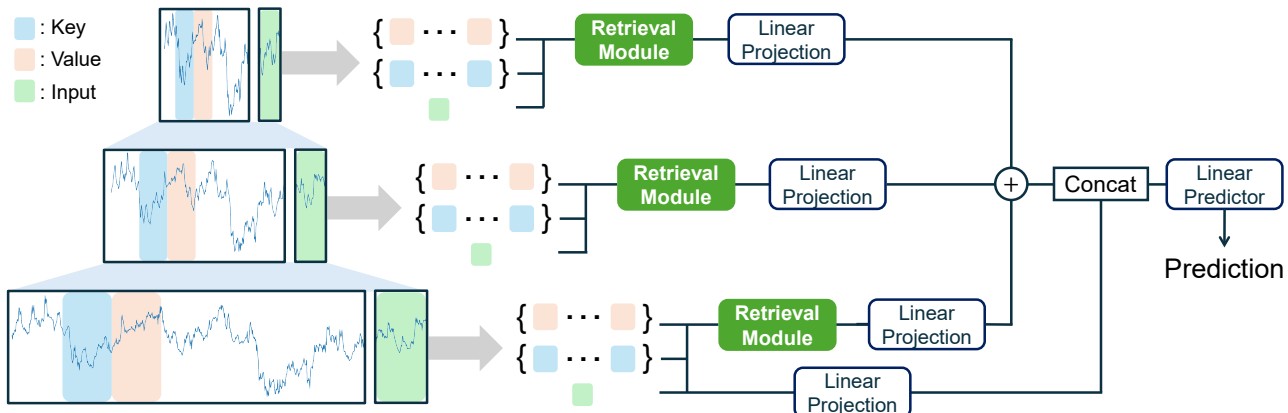

*Figure 3.* Illustration of the proposed architecture, RAFT. The input time series **x** and the entire past observed time series **S** are first downsampled to generate multiple series with different periods. Then, a retrieval module is applied to each series to retrieve information relevant to the current input. The retrieved results are projected to the same dimension via a linear layer, and the results from different periods are summed to aggregate the information. Finally, the input time series is concatenated with the aggregated retrieved results, and a linear layer is applied to produce the final prediction.

Figure 3 illustrates our model's forecasting process with multiple periods of retrieval. Hyperparameters such as $m$ are chosen based on the performance in the validation set.

## 4. Experiments

We evaluate RAFT across multiple time series forecasting benchmark datasets. We analyze how our proposed retrieval module contributes to performance improvement in time-series forecasting, and in which scenarios retrieval is particularly beneficial. The full results, visualizations, and additional analyses of our model are provided in the Appendix.

### 4.1. Experimental settings

**Datasets.** We consider ten different benchmark datasets, each with a diverse range of variates, dataset lengths, and frequencies: (1-4) The ETT dataset contains 2 years of electricity transformer temperature data, divided into four subsets—ETTh1, ETTh2, ETTm1, and ETTm2 (Zhou et al., 2021); (5) The Electricity dataset records household electric power consumption over approximately 4 years (Trindade, 2015); (6) The Exchange dataset includes the daily exchange rates of eight countries over 27 years (1990–2016) (Lai et al., 2018); (7) The Illness dataset includes the weekly ratio of patients with influenza-like illness over 20 years (2002-2021)[4]; (8) The Solar dataset contains 10-minute solar power forecasts collected from power plants in 2006 (Liu et al., 2022a); (9) The Traffic dataset contains hourly road occupancy rates on freeways over 48 months[5]; and (10) The Weather dataset

consists of 21 weather-related indicators in Germany over one year[6]. Data summary is provided in the Appendix A.

**Baselines.** We compare against 9 contemporary time-series forecasting baselines, including: (1) Autoformer (Wu et al., 2021), (2) Informer (Zhou et al., 2021), (3) Stationary (Liu et al., 2022b), (4) Fedformer (Zhou et al., 2022), and (5) PatchTST (Nie et al., 2023), all of which use Transformer-based architectures; (6) DLinear (Zeng et al., 2023), which are lightweight models with simple linear architectures; (7) MICN (Wang et al., 2023), which leverages both local features and global correlations through a convolutional structure; (8) TimesNet (Wu et al., 2023), which utilizes Fourier Transformation to decompose time-series data within a modular architecture; and (9) TimeMixer (Wang et al., 2024), which utilizes decomposition and multi-periodicity for forecasting[7].

**Implementation details.** RAFT employs the retrieval module with following detailed settings. The periods are set to $\{1, 2, 4\}$ $(n = 3)$, following existing literature (Wang et al., 2024), and the temperature $\tau$ is set to 0.1. Batch size is set to 32. The initial learning rate, the number of patches used in the retrieval $(m)$, and the size of the look-back window $(L)$ are determined via grid search based on performance on the validation set, following the prior work (Wang et al., 2024). For fair comparison, hyper-parameter tuning was performed for both our model and all baselines using the validation set. The learning rate is chosen from 1e-5 to 0.05, look back

---

[4]https://gis.cdc.gov/grasp/fluview/fluportaldashboard.html
[5]https://pems.dot.ca.gov/

[6]https://www.bgc-jena.mpg.de/wetter/
[7]We compare our model with general time-series forecasting models. Other retrieval-based time-series models mentioned in the related work assume the presence of multiple time-series instances, which are outside the scope of our study.

*Table 1.* Comparison of RAFT and baseline methods across 10 datasets using MSE. For all datasets except Illness, results are averaged over forecasting horizons of 96, 192, 336, and 720. For the Illness dataset, forecasting horizons of 24, 36, 48, and 60 are used. Best performances are bolded, and our framework's performances, when second-best, are underlined.

| Methods | RAFT | TimeMixer | PatchTST | TimesNet | MICN | DLinear | FEDformer | Stationary | Autoformer | Informer |
|---|---|---|---|---|---|---|---|---|---|---|
| ETTh1 | **0.420** | 0.447 | 0.516 | 0.495 | 0.475 | 0.461 | 0.498 | 0.570 | 0.496 | 1.040 |
| ETTh2 | **0.359** | 0.364 | 0.391 | 0.414 | 0.574 | 0.563 | 0.437 | 0.526 | 0.450 | 4.431 |
| ETTm1 | **0.348** | 0.381 | 0.406 | 0.400 | 0.423 | 0.404 | 0.448 | 0.481 | 0.588 | 0.961 |
| ETTm2 | **0.254** | 0.275 | 0.290 | 0.291 | 0.353 | 0.354 | 0.305 | 0.306 | 0.327 | 1.410 |
| Electricity | **0.160** | 0.182 | 0.216 | 0.193 | 0.196 | 0.225 | 0.214 | 0.193 | 0.227 | 0.311 |
| Exchange | 0.441 | **0.386** | 0.564 | 0.416 | 0.315 | 0.643 | 1.195 | 0.461 | 1.447 | 2.478 |
| Illness | 2.097 | 2.024 | **1.480** | 2.139 | 2.664 | 2.169 | 2.847 | 2.077 | 3.006 | 5.137 |
| Solar | 0.231 | **0.216** | 0.287 | 0.403 | 0.283 | 0.330 | 0.328 | 0.350 | 0.586 | 0.331 |
| Traffic | **0.434** | 0.484 | 0.529 | 0.620 | 0.593 | 0.625 | 0.610 | 0.624 | 0.628 | 0.764 |
| Weather | 0.241 | **0.240** | 0.265 | 0.251 | 0.268 | 0.265 | 0.309 | 0.288 | 0.338 | 0.634 |

window size from $\{96, 192, 336, 720\}$, and the number of patches used in retrieval $m$ from $\{1, 5, 10, 20\}$. The chosen values of each setting are presented in the Appendix B. For implementation, we referred to the publicly available time-series repository (TSLib)[8]. For all experiments, the average results from three runs are reported, with each experiment conducted on a single NVIDIA A100 40GB GPU. For more details about the computational complexity analysis of RAFT, see Appendix D.

**Evaluation.** We consider two metrics for evaluation: MSE and MAE. We varied the forecasting horizon length to measure performance (i.e., $F = 96, 192, 336, 720$), and each experiment setting was run with three different random seeds to compute the average results. For the Illness dataset, forecasting horizons of 24, 36, 48, and 60 are used, following the prior work (Nie et al., 2023; Wang et al., 2024). The evaluation was conducted in multivariate settings, where both the input and forecasting target have multiple channels.

### 4.2. Experimental results on forecasting benchmarks

Table 1 presents comparisons between the performance of time series forecasting methods and RAFT. The results represent the average MSE performance evaluated across different forecasting horizon lengths. We observe that our model consistently outperforms other contemporary baselines on average, supporting the effectiveness of retrieval in time series forecasting. Full results and comparisons using a different evaluation metric (i.e., MAE) are provided in Appendix G.

## 5. Discussions

In this section, we explore scenarios where retrieval shows substantial advantage by empirically analyzing its effect,

using both benchmark time series datasets and synthetic time series datasets.

### 5.1. Better retrieval results lead to better performance.

Two criteria are important for our retrieval method to enhance the forecasting performance. First, the value patches $\mathcal{V}$ identified through the similarity between the input query $\mathbf{x}$ and key patches $\mathcal{K}$ should closely match the actual future value $\mathbf{y}_0$ which sequentially follows the input query. Second, the model should efficiently leverage the information in the value patches for forecasting. From these, we can draw the insight that higher similarity between input query and key patches (i.e., key similarity) will lead to the higher similarity between the actual value and value patches (i.e., value similarity), eventually resulting in better performance.

Figure 4 presents the correlation analysis conducted on the ETTh1 dataset. Figure 4a shows that retrieving key patches with higher similarity leads to value patches that are more closely aligned with the actual future value. Figure 4b illustrates that the value patches with greater similarity to the actual future values tend to improve RAFT's performance more significantly. This trend is also consistent across datasets; datasets with higher key similarity show higher value similarity, resulting in larger performance gains. Spearman's correlation coefficient validate this trend, showing a correlation of $0.60$ between key similarity and value similarity, and a correlation of $-0.54$ between value similarity and performance gain across datasets. The negative correlation with performance is due to the use of MSE as the metric (lower the better). These results demonstrate that better retrieval results from the retrieval module lead to improved performance of RAFT.

### 5.2. Retrieval is helpful when rare patterns repeat.

RAFT can complement scenarios where a particular pattern does not frequently appear in the training dataset, making it

---

[8]https://github.com/thuml/
Time-Series-Library

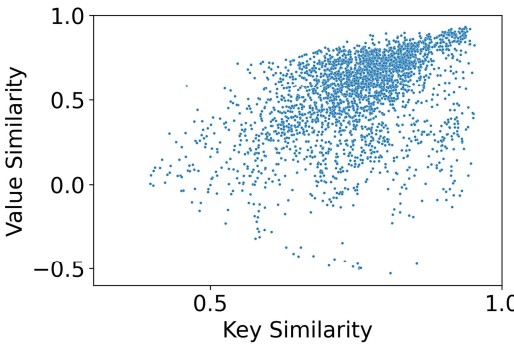

(a) Scatter plot of key and value similarity

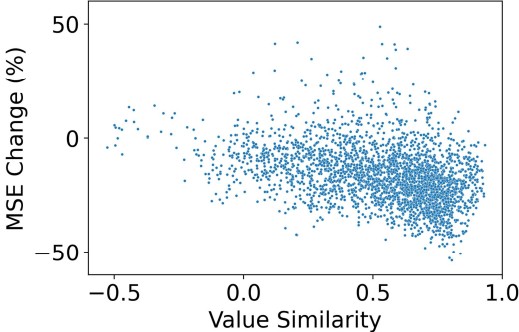

(b) Scatter plot of value similarity and MSE change (%)

*Figure 4.* Analysis of the correlation between (a) the key similarity and value similarity, and (b) the value similarity and model performance changes measured by MSE (%). Each dot represents each input patch from the ETTh1 test dataset. Key similarity refers to the average similarity between input query ($\mathbf{x}$) and all retrieved key patches ($\mathcal{K}$). Value similarity refers to the average similarity between actual future value ($\mathbf{y}_0$) and all retrieved value patches ($\mathcal{V}$).

difficult for the model to memorize. By utilizing retrieved information, the model can overcome this challenge. To analyze this effect, we conducted experiments using synthetic time series datasets.

**Synthetic data generation with autoregressive model.** The synthetic time series was constructed by combining three components: trend, seasonality, and event-based short-term patterns. Trend and seasonality were generated using sinusoidal functions with varying periods, amplitudes, and offsets, representing long-term consistent patterns. Short-term patterns, modeled as event-based dynamics, were created using an autoregressive model:

$$x_t = \sum_{i=1}^{20} \varphi_i x_{t-i} + \epsilon_t, \tag{10}$$

where $\varphi_i$ are autoregressive parameters, and $\epsilon_t$ is noise sampled from a uniform distribution. The short-term pattern length was fixed at 200. To test retrieval effectiveness for rare patterns, we generated three distinct short-term patterns and varied their frequency in the training dataset. Forecasting accuracy (MSE) was evaluated when each short-term pattern appeared in the test set, with input and forecasting horizon lengths fixed at 96. Additional dataset details and figures are available in Figure 5a and Appendix E.

**Results.** Table 2 presents the number of occurrences of the short-term patterns and the corresponding performance of RAFT with and without retrieval, as well as baseline models. Note that, in this experiment, we did not consider multiple periods in order to isolate the effect of retrieval, so RAFT without retrieval has an identical structure to the NLinear (Zeng et al., 2023). The results show that our model, utilizing retrieval, consistently outperformed the model without retrieval on the synthetic dataset; 9.2∼14.7% increase in performance depending on the pattern occurrences. Notably, as the pattern occurrences decreased, the

*Table 2.* Analysis between forecasting accuracy and the rarity of the pattern over the synthetic time series with an autoregressive model. Forecasting accuracy was evaluated using MSE, averaged across 120 different time series and short-term patterns. The numbers in the last row indicate the ratio by which the MSE decreases when retrieval is appended.

| Pattern occurrences | 1 | 2 | 4 |
|---|---|---|---|
| TimeMixer | 0.2360 | 0.2166 | 0.2276 |
| TimesNet | 0.2282 | 0.1970 | 0.1925 |
| MICN | 0.2285 | 0.2331 | 0.2033 |
| DLinear | 0.2640 | 0.2552 | 0.2502 |
| RAFT without Retrieval | 0.2590 | 0.2310 | 0.2344 |
| RAFT with Retrieval | 0.2209 | 0.2064 | 0.2128 |
| MSE decrease ratio | -14.7% | -10.7% | -9.2% |

reduction in MSE was more significant. Similar to RAFT without retrieval, the baseline models exhibited a decrease in performance as the pattern occurrences decreased. When we also visualize the predictions of models with and without retrieval modules over the rare pattern (see Figure 5b), the model utilizing retrieval aligns well with the pattern's periodicity and offset during forecasting, while the model relying solely on learning fails to capture these aspects. This suggests that the model struggles to learn rare patterns, and the retrieval module effectively complements this deficiency.

### 5.3. Retrieval is helpful when patterns are temporally less correlated.

If short-term patterns are very similar across time, there's less unique information for the model to learn, making it easier to achieve accurate predictions. On the other hand, if the short-term patterns in time series data are similar to a random walk without any specific temporal correlation, the model would need to memorize all changes within

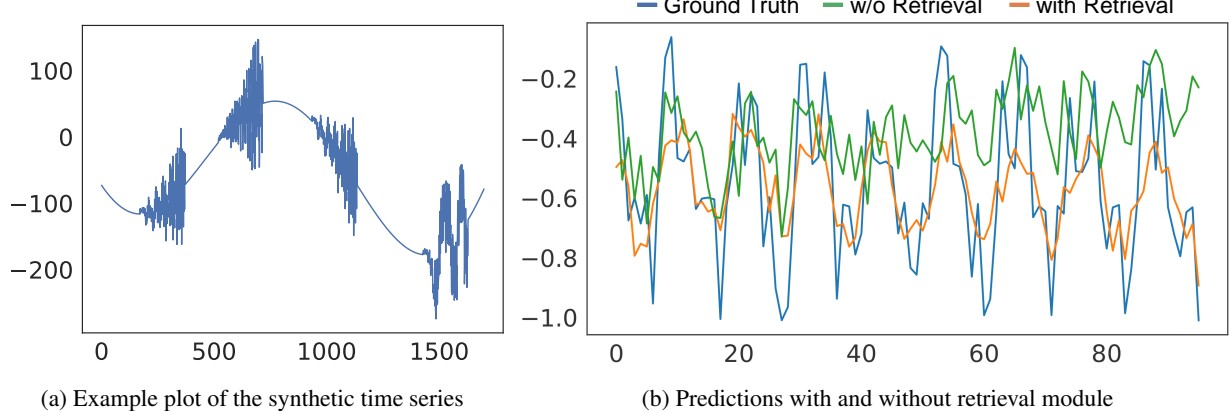

(a) Example plot of the synthetic time series        (b) Predictions with and without retrieval module

*Figure 5.* Visualization of a synthetic time series with short-term patterns and the corresponding predictions over the rare short-term pattern from models with and without the retrieval module. MSE of predictions in this example without retrieval is 0.087, while with retrieval, it improves to 0.035.

short-term pattern for accurate forecasting. Based on this hypothesis, we expect the retrieval module to be especially helpful when patterns are temporally less correlated, as retrieval can easily detect similarities between patterns that temporal correlation alone cannot capture. We again use the synthetic dataset for validation.

**Synthetic data generation with random walk model.** Instead of generating short-term patterns using the autoregressive model as before, we utilize random walk-based change patterns, following the equation:

$$x_t = x_{t-1} + \epsilon_t. \tag{11}$$

The step size for the walk $\epsilon_t$ was sampled from a uniform distribution within the range of [-20, 20]. The generated short-term patterns were then inserted into the training data, as in the previous synthetic time-series approach.

*Table 3.* Forecasting accuracy over the rarity of the pattern. Synthetic time series with random walk based patterns (temporally less correlated) is used. Forecasting accuracy was evaluated using MSE, averaged across 120 different time series and short-term patterns. The numbers in the last row indicate the ratio by which the MSE decreases when retrieval is appended.

| Pattern occurrences | 1 | 2 | 4 |
|---|---|---|---|
| TimeMixer | 0.2863 | 0.2305 | 0.2249 |
| TimeNet | 0.2448 | 0.1877 | 0.1938 |
| MICN | 0.2536 | 0.2445 | 0.2450 |
| DLinear | 0.3175 | 0.2059 | 0.2798 |
| RAFT without retrieval | 0.2694 | 0.2649 | 0.1894 |
| RAFT with retrieval | 0.1845 | 0.1818 | 0.1592 |
| MSE decrease ratio | -31.5% | -31.4% | -16.0% |

**Results.** Table 3 shows the results of applying the same experiment as in Table 2, but with different synthetic time-

series data. Again, the retrieval module improves performance across all cases, particularly for rare patterns. Furthermore, the performance improvement is more significant for temporally less correlated patterns (16.0∼31.5% decrease of MSE depending on pattern occurrences), compared to temporally more correlated ones shown in Table 2 (9.2∼14.7%). The baseline models exhibited a similar trend to that observed in Table 2, while the performance gap compared to RAFT with retrieval has become more significant. This confirms that the proposed retrieval module is more beneficial when dealing with temporally less correlated or near-random patterns that are more challenging for the model to learn.

### 5.4. Retrieval is also helpful for Transformer-variants

We investigate the effectiveness of the retrieval module Transformer-variants, using AutoFormer. Instead of modifying the internal Transformer architecture to integrate our retrieval module, we directly added retrieval results to AutoFormer's predictions at the final stage. Table 4 demonstrates that our retrieval module successfully enhances the forecasting performance of the Transformer-based model, highlighting its broader applicability to other architectures.

*Table 4.* Performance comparison between AutoFormer and AutoFormer with our proposed retrieval module. The average MSE across different forecasting horizon lengths is reported.

|  | ETTh1 | ETTh2 | ETTm1 | ETTm2 |
|---|---|---|---|---|
| Autoformer | 0.496 | 0.450 | 0.588 | 0.327 |
| + Retrieval | **0.471** | **0.444** | **0.454** | **0.326** |

## 6. Conclusion

In this paper, we introduce RAFT, a time-series forecasting method that leverages retrieval from training data to augment the input. Our retrieval module lessens the model to absorb all unique patterns in its weights, particularly those that lack temporal correlation or do not share common characteristics with other patterns. This overall is demonstrated as an effective inductive bias for deep learning architectures for time-series. Our extensive evaluations on numerous real-world and synthetic datasets confirm that RAFT achieves performance improvements over contemporary baselines. As various retrieval-based models are being proposed, there remains room for improvement in retrieval techniques specifically tailored for time-series data (beyond the simple approaches used), including determining when, where, and how to apply retrieval based on dataset characteristics and capture more complex similarity measures that depend on nonlinear and nonstationary characteristics. Our work is expected to open new avenues in the time-series forecasting field through the use of retrieval-augmented approaches.

## Acknowledgement

We would like to thank Tomas Pfister and Dhruv Madeka for their valuable feedback during the review of our paper. Han, Lee, and Cha were partially supported by the National Research Foundation of Korea grant (RS-2022-00165347). Han was partially supported by GIST (AI-based Research Scientist Project).

## Impact Statement

This paper advances time series forecasting by enhancing deep learning models with the retrieval of information from historical training data. By reducing the learning burden and mitigating memorization, this approach improves the forecasting performance in various real-world applications, such as weather prediction and financial analysis. We believe this work offers a new perspective on integrating the concept of retrieval into the time-series domain.

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

# A. Dataset Details

In this work, we use widely-used 10 time series datasets. The detailed information of each dataset are shown in Table 5. The dataset size is presented in (Train, Validation, Test). The targets used in the univariate setting are as follows: oil temperature for the ETTh1, ETTh2, ETTm1, ETTm2 datasets; the consumption of a client for the Electricity dataset; the exchange rate of Singapore for the Exchange Rate dataset; the weekly ratio of patients for Illness dataset; 10-minute solar power forecasts collected from power plants for the Solar dataset; the road occupancy rates measured by a sensor for the Traffic dataset; and $CO_2$ (ppm) for the Weather dataset.

*Table 5.* Basic information of datasets used for evaluation.

| Dataset | # of variates | Dataset Size | Frequency |
|---|---|---|---|
| ETTh1 | 7 | (8449, 2785, 2785) | Hourly |
| ETTh2 | 7 | (8449, 2785, 2785) | Hourly |
| ETTm1 | 7 | (34369, 11425, 11425) | 15 min |
| ETTm2 | 7 | (34369, 11425, 11425) | 15 min |
| Electricity | 321 | (18221, 2537, 5165) | Hourly |
| Exchange Rate | 8 | (5120, 665, 1422) | Daily |
| Illness | 7 | (485, 2, 98) | Weekly |
| Solar | 137 | (36601, 5161, 10417) | 10 min |
| Traffic | 862 | (12089, 1661, 3413) | Hourly |
| Weather | 21 | (36696, 5175, 10444) | 10min |

## B. Implementation Details

RAFT employs a retrieval module with the following detailed settings. The periods are set to $1, 2, 4$ ($n = 3$), following existing literature (Wang et al., 2024). The temperature $\tau$ is set to 0.1. The remaining settings, including the look back window size $L$, the learning rate, and the number of patches used in retrieval $m$ are determined through grid search based on validation set performance, consistent with prior work (Wang et al., 2024). The effect of hyper-parameters ($L, m, \tau$) on the performance are analyzed in the Section C.3-C.4.

Table 6 provides the parameter settings of our model for each dataset. We observed that some parameters vary across different datasets.

*Table 6.* The chosen parameter values of each setting via grid search over the validation set.

|  | Forecasting horizon size | Look back window size | Learning rate | Number of retrievals |
|---|---|---|---|---|
| ETTh1 | 96 | 720 | 1.00E-03 | 20 |
|  | 192 | 720 | 1.00E-02 | 20 |
|  | 336 | 720 | 1.00E-02 | 20 |
|  | 720 | 720 | 1.00E-04 | 20 |
| ETTh2 | 96 | 720 | 1.00E-02 | 10 |
|  | 192 | 720 | 1.00E-03 | 10 |
|  | 336 | 720 | 1.00E-03 | 20 |
|  | 720 | 720 | 1.00E-04 | 20 |
| ETTm1 | 96 | 720 | 1.00E-02 | 1 |
|  | 192 | 720 | 1.00E-03 | 20 |
|  | 336 | 720 | 1.00E-03 | 20 |
|  | 720 | 720 | 1.00E-02 | 20 |
| ETTm2 | 96 | 720 | 1.00E-03 | 5 |
|  | 192 | 720 | 1.00E-03 | 20 |
|  | 336 | 720 | 1.00E-04 | 20 |
|  | 720 | 720 | 1.00E-04 | 20 |
| Electricity | 96 | 720 | 1.00E-02 | 1 |
|  | 192 | 720 | 1.00E-03 | 1 |
|  | 336 | 720 | 1.00E-03 | 1 |
|  | 720 | 720 | 1.00E-03 | 1 |
| Exchange | 96 | 720 | 1.00E-04 | 1 |
|  | 192 | 720 | 1.00E-03 | 1 |
|  | 336 | 720 | 1.00E-03 | 10 |
|  | 720 | 720 | 1.00E-04 | 20 |
| Illness | 96 | 96 | 1.00E-02 | 1 |
|  | 192 | 96 | 1.00E-02 | 1 |
|  | 336 | 96 | 1.00E-02 | 20 |
|  | 720 | 96 | 1.00E-02 | 20 |
| Solar | 96 | 720 | 1.00E-03 | 1 |
|  | 192 | 720 | 1.00E-02 | 1 |
|  | 336 | 720 | 1.00E-03 | 1 |
|  | 720 | 720 | 1.00E-03 | 1 |
| Traffic | 96 | 720 | 1.00E-02 | 1 |
|  | 192 | 720 | 1.00E-03 | 1 |
|  | 336 | 720 | 1.00E-03 | 1 |
|  | 720 | 720 | 1.00E-03 | 1 |
| Weather | 96 | 720 | 1.00E-02 | 1 |
|  | 192 | 720 | 1.00E-03 | 1 |
|  | 336 | 720 | 1.00E-03 | 1 |
|  | 720 | 720 | 1.00E-03 | 1 |

## C. Component Analysis

In this section, we analyze the impact of each component of RAFT on performance.

### C.1. Different Similarity Metrics for Retrieval

We compared RAFT using various similarity metrics, including Pearson's correlation, cosine similarity, cosine similarity with projection, and negative L2 distance. Cosine similarity with projection employs a trainable linear projection head for the input query and key vectors, respectively, and measures cosine similarity between the embeddings after projection rather than between the raw query and key. Table 7 presents the comparison results across datasets, where Pearson's correlation shows the best performance among the various similarity metrics. We also observe that the linear projection provide a comparable performance to measuring similarity with the raw query and key.

Beyond performance, this projection-based approach offers computational advantages. By mapping sparse, high-dimensional inputs into dense, lower-dimensional embeddings, it reduces the time complexity of retrieval - especially when integrated with preprocessing, parallelization, and efficient vector search techniques such as approximate nearest neighbor (ANN) methods. Additionally, the projection mechanism enables incorporation of external features or correlated time series. In such scenarios, separate encoders can project external segments and input features into a shared embedding space. Once aligned, cosine similarity can be used to retrieve relevant external segments for each input query. These retrieved segments can then be leveraged alongside the original input to enhance predictive performance, with both encoders optimized jointly during training.

*Table 7.* Comparison of various similarity metrics with RAFT in the univariate setting.

|  | Pearson's Correlation | Cosine Similarity | Cosine Sim with Projection | Negative L2 Distance |
|---|---|---|---|---|
| ETTh1 | **0.0559** | 0.0561 | 0.0562 | 0.0562 |
| ETTh2 | **0.1231** | 0.1235 | 0.1298 | 0.1271 |
| ETTm1 | 0.0299 | 0.0298 | **0.0294** | 0.0296 |
| ETTm2 | **0.0647** | 0.0649 | 0.0699 | 0.0666 |
| Electricity | **0.3307** | 0.3343 | 0.3981 | 0.3388 |
| Exchange Rate | **0.0915** | 0.0917 | 0.0933 | 0.0922 |
| Traffic | **0.2737** | 0.2773 | 0.2943 | 0.2925 |
| Weather | 0.0118 | 0.0129 | **0.0026** | 0.0278 |

### C.2. Ablation Study on Retrieval Module

To thoroughly analyze the impact of the proposed retrieval design on performance, we conducted an ablation study on the retrieval module. The ablations were as follows: (1) Random Retrieval – Key patches are retrieved randomly, without considering similarity to the query; (2) Without Attention – When aggregating value patches, we use equal weights instead of similarity-based weights (Eq. 5); (3) With One Period – Only period 1 is used for retrieval ($\mathcal{P} = \{1\}$); (4) Without Retrieval – Retrieval is entirely removed, leaving only the linear predictor. The experiments were conducted under identical hyper-parameter and learning settings and evaluated on multivariate forecasting tasks. Table 8 presents the MSE results for each dataset across the ablations. As shown in the results, our model with all components included consistently achieved the best performance compared to the baselines across all datasets. Notably, we observed that when retrieval was conducted randomly, without attention or with one period, performance was sometimes even worse than without retrieval, which demonstrates that retrieving relevant data is crucial for achieving high performance.

*Table 8.* Ablation study on retrieval module in the multivariate setting.

|  | ETTh1 | ETTh2 | ETTm1 | ETTm2 | Electricity | Exchange Rate | Traffic | Weather |
|---|---|---|---|---|---|---|---|---|
| RAFT | **0.367** | **0.276** | **0.302** | **0.164** | **0.133** | **0.091** | **0.378** | **0.165** |
| Random Retrieval | 0.382 | 0.282 | 0.305 | 0.171 | 0.150 | 0.092 | 0.413 | 0.188 |
| Without Attention | 0.379 | 0.281 | 0.300 | 0.165 | 0.148 | 0.090 | 0.409 | 0.172 |
| With One Period | 0.369 | 0.276 | 0.303 | 0.164 | 0.133 | 0.088 | 0.379 | 0.168 |
| Without Retrieval | 0.379 | 0.282 | 0.306 | 0.167 | 0.143 | 0.089 | 0.410 | 0.182 |

## C.3. Effect of Look Back Window Size ($L$)

We analyze the effect of look back window size ($L$) on forecasting performance. Keeping all other experimental settings fixed, we varied the look back window size between 96, 192, 336, and 720 to observe performance changes. The experiments were conducted in a multivariate setting across four datasets, with the prediction length set to 96. Table 9 compares the MSE results for different look back window sizes. Consistent with prior works (Wang et al., 2024; Zeng et al., 2023), we observed that RAFT, based on a linear model, also achieves better forecasting performance as the look back window size increases.

*Table 9.* Comparison results over different look back window size.

| Look back window size ($L$) | 96 | 192 | 336 | 720 |
|---|---|---|---|---|
| ETTh1 | 0.387 | 0.390 | 0.386 | 0.367 |
| ETTh2 | 0.296 | 0.292 | 0.281 | 0.276 |
| ETTm1 | 0.348 | 0.310 | 0.306 | 0.302 |
| ETTm2 | 0.179 | 0.171 | 0.166 | 0.164 |

## C.4. Hyper-Parameter Analysis

RAFT has two key internal model parameters. The first is the number of patches retrieved by the retrieval module, and the second is the temperature $\tau$ used in the softmax function to calculate weights. Each hyper-parameter is optimally tuned for each dataset based on the validation set. Table 10-11 below illustrates examples of performance variations (MSE) across four datasets with different hyper-parameter values. As shown, the optimal values of the hyper-parameters vary depending on the dataset.

*Table 10.* Effect of the number of retrievals ($m$) on performance.

| The number of retrievals ($m$) | 1 | 5 | 10 | 20 |
|---|---|---|---|---|
| ETTh1 | 0.370 | 0.368 | 0.367 | 0.367 |
| ETTh2 | 0.280 | 0.278 | 0.276 | 0.275 |
| ETTm1 | 0.302 | 0.300 | 0.298 | 0.297 |
| ETTm2 | 0.164 | 0.164 | 0.164 | 0.164 |

*Table 11.* Effect of the temperature ($\tau$) on performance.

| Temperature ($\tau$) | 0.01 | 0.1 | 1 | 10 |
|---|---|---|---|---|
| ETTh1 | 0.383 | 0.367 | 0.378 | 0.381 |
| ETTh2 | 0.285 | 0.276 | 0.280 | 0.281 |
| ETTm1 | 0.303 | 0.302 | 0.300 | 0.304 |
| ETTm2 | 0.165 | 0.164 | 0.165 | 0.167 |

## C.5. Effect of Training Time Series Length

When the available time series is short, the number of historical patches for retrieval naturally decreases, potentially limiting performance gains. To investigate this phenomenon, we conducted additional experiments comparing our model with and without retrieval, varying the length of the training data from our benchmark datasets and evaluated performance on a fixed test set while adjusting the training data proportion to 20%, 60%, and 100%. The results presented in Table 12 indicate that the impact of the training dataset's length on performance gains varies significantly across different datasets. However, our approach consistently outperformed the baseline model, even with limited historical data.

*Table 12.* Effect of training time series length on RAFT performance.

| Dataset | Training data proportion | 20% | 60% | 100% |
|---------|--------------------------|-------|-------|-------|
| ETTh1 | RAFT without retrieval | 0.590 | 0.444 | 0.379 |
|  | RAFT with retrieval | 0.562 | 0.428 | 0.367 |
| ETTh2 | RAFT without retrieval | 0.251 | 0.255 | 0.282 |
|  | RAFT with retrieval | 0.260 | 0.255 | 0.276 |
| ETTm1 | RAFT without retrieval | 1.492 | 0.692 | 0.306 |
|  | RAFT with retrieval | 0.975 | 0.684 | 0.302 |
| ETTm2 | RAFT without retrieval | 1.364 | 0.608 | 0.167 |
|  | RAFT with retrieval | 1.312 | 0.603 | 0.164 |

## D. Computational Complexity for Retrieval

Our model incorporates a retrieval process to find similar patches in the given data. For efficient training, the retrieval process is pre-computed for the training and validation data, requiring computation only once during training. We analyzed the wall time (in seconds) for retrieval pre-computation, training, and inference on the ETTm1 dataset (see Table 13). The lookback window size was set to 720, and the forecasting horizon length was set to 96.

*Table 13.* Wall time for each process of RAFT over ETTm1.

|  | Pre-computation | Training time per epoch | Total Inference time |
|---|---|---|---|
| Wall time (sec) | 42.2 | 7.3 | 1.9 |

The pre-computation speed for retrieval of our model is $O(N^2)$, where $N$ denotes the size of the time-series in the training data. To reduce this time, one approach is to increase the stride of the sliding window beyond 1, speeding up the search process. Table 14 records the changes in wall time as the stride of the sliding window increases. As the stride increases, the time required for the search process decreases significantly.

*Table 14.* Wall time across different number of strides over ETTm1.

| Stride | 1 | 2 | 4 | 8 |
|---|---|---|---|---|
| Wall time for pre-computation (sec) | 42.2 | 19.8 | 9.3 | 4.7 |

Lastly, we examined the impact of increasing the stride on forecasting performance. Table 15 presents the changes in MSE across four datasets (ETTh1, ETTh2, ETTm1, ETTm2) as the stride increases. While increasing the stride introduced a performance trade-off, we observed that the decrease in performance was not significant.

*Table 15.* MSE changes of RAFT over four datasets across the different number of strides.

| Stride | 1 | 2 | 4 | 8 |
|---|---|---|---|---|
| ETTh1 | 0.367 | 0.379 | 0.381 | 0.383 |
| ETTh2 | 0.276 | 0.279 | 0.279 | 0.280 |
| ETTm1 | 0.302 | 0.298 | 0.299 | 0.300 |
| ETTm2 | 0.164 | 0.164 | 0.165 | 0.165 |

# E. Synthetic Dataset Generation Details

The synthetic time series was created by combining three different components. Two of these components represent trend and seasonality, which exhibit long-term consistent patterns throughout the entire time series. The third component represents event-based short-term patterns. The generation details for each component are as follows:

**Trend and seasonality components.** To generate the trend and seasonality components, we synthesized sinusoidal functions with varying periods, amplitudes, and offsets. The total length of the time series was set to 18,000. The period of the sinusoidal function for the trend was sampled from a uniform distribution between [1000, 4000], while the period for seasonality was shorter, sampled from [500, 1000]. The amplitude of each component was randomly chosen from the ranges [200, 300] for the trend and [100, 200] for the seasonality. Offsets were sampled from the range [100, 200].

**Short-term patterns from the autoregressive model.** The length of each short-term pattern was set to 200. In the case of the autoregressive model, the value of the next time step was determined by the previous 20 time steps, following the equation below:

$$x_t = \sum_{i=1}^{20} \varphi_i x_{t-i} + \epsilon_t, \tag{12}$$

where $\varphi_i$ represents the parameters in the autoregressive model, and $\epsilon_t$ is the noise. The parameters were sampled from a uniform distribution within [-5, 5], and the noise was sampled from a uniform distribution within [-10, 10]. The length of the short-term pattern was set to 200. To prevent the short-term patterns from producing extreme values compared to the trend and seasonal components, we clamped the values within the range [-100, 100].

**Short-term patterns from the random-walk model.** In the case of the random-walk model, the length of the short-term pattern was also fixed at 200. Unlike the autoregressive model, in the random-walk model, the value of the next time step depends only on the previous time step, as described by the equation:

$$x_t = x_{t-1} + \epsilon_t., \tag{13}$$

where the step size for the walk was sampled from a uniform distribution within the range of [0, 20]. Again, to prevent the short-term patterns from producing extreme values compared to the trend and seasonal components, we clamped the values within the range [-100, 100].

Finally, the trend, seasonality, and short-term patterns were combined to create the synthetic time series. Example visualizations of the autoregressive short-term pattern, the random-walk pattern, and the resulting synthetic time series can be seen in Figure 6.

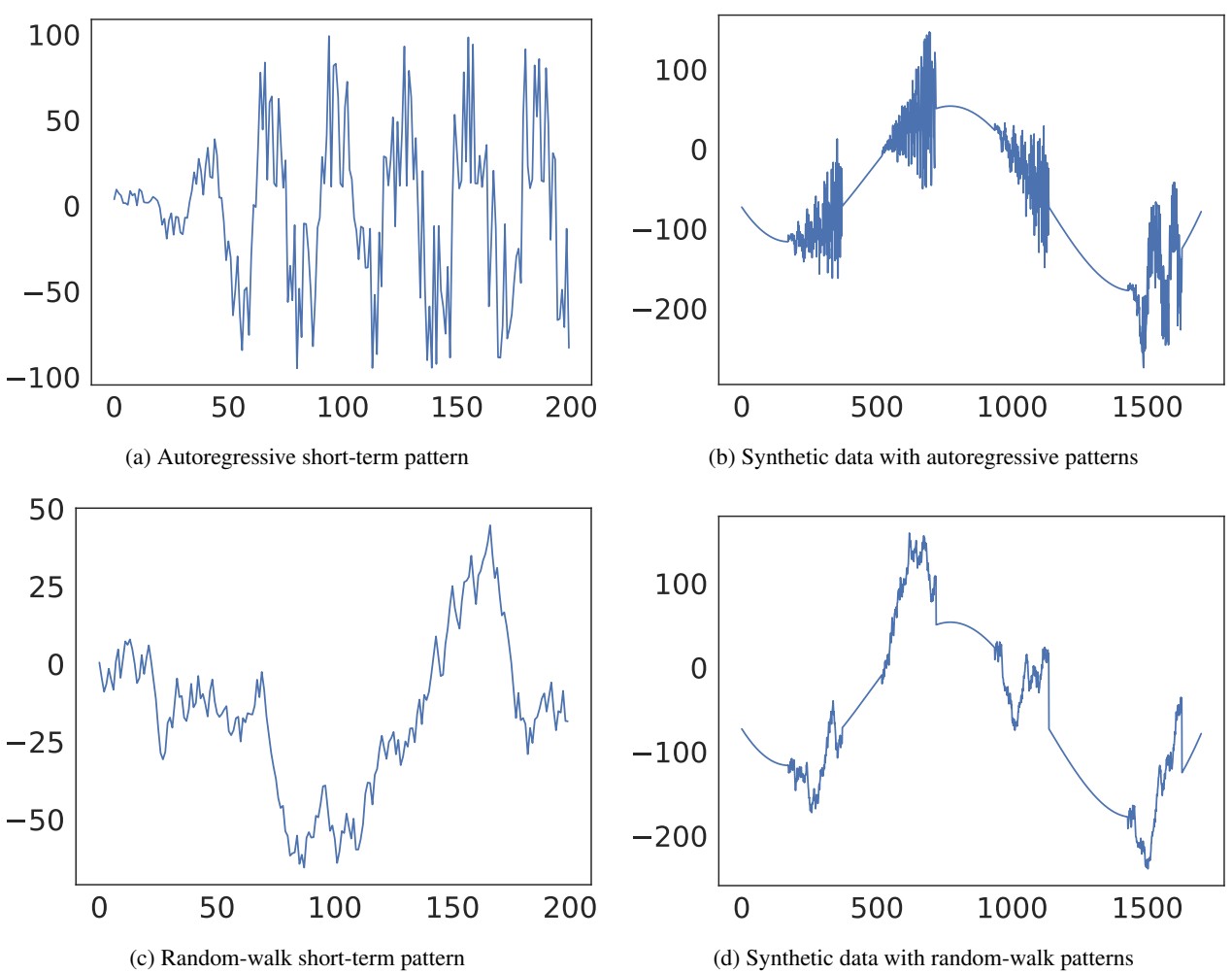

(a) Autoregressive short-term pattern

(b) Synthetic data with autoregressive patterns

(c) Random-walk short-term pattern

(d) Synthetic data with random-walk patterns

*Figure 6.* Visualization of an example synthetic time series with short-term patterns.

# F. Qualitative Analysis on Retrieval

In this section, we provide examples of our retrieval results. Figure 7-9 illustrate a comparison between the input query and the retrieved key patch, as well as a comparison between the ground truth and the retrieved value patch, with retrievals by 1, 2, and 4 periods. Note that we retrieve the key patch with the top-1 similarity and its following value patch. The results demonstrate that our retrieval module effectively delivers useful information for forecasting future predictions.

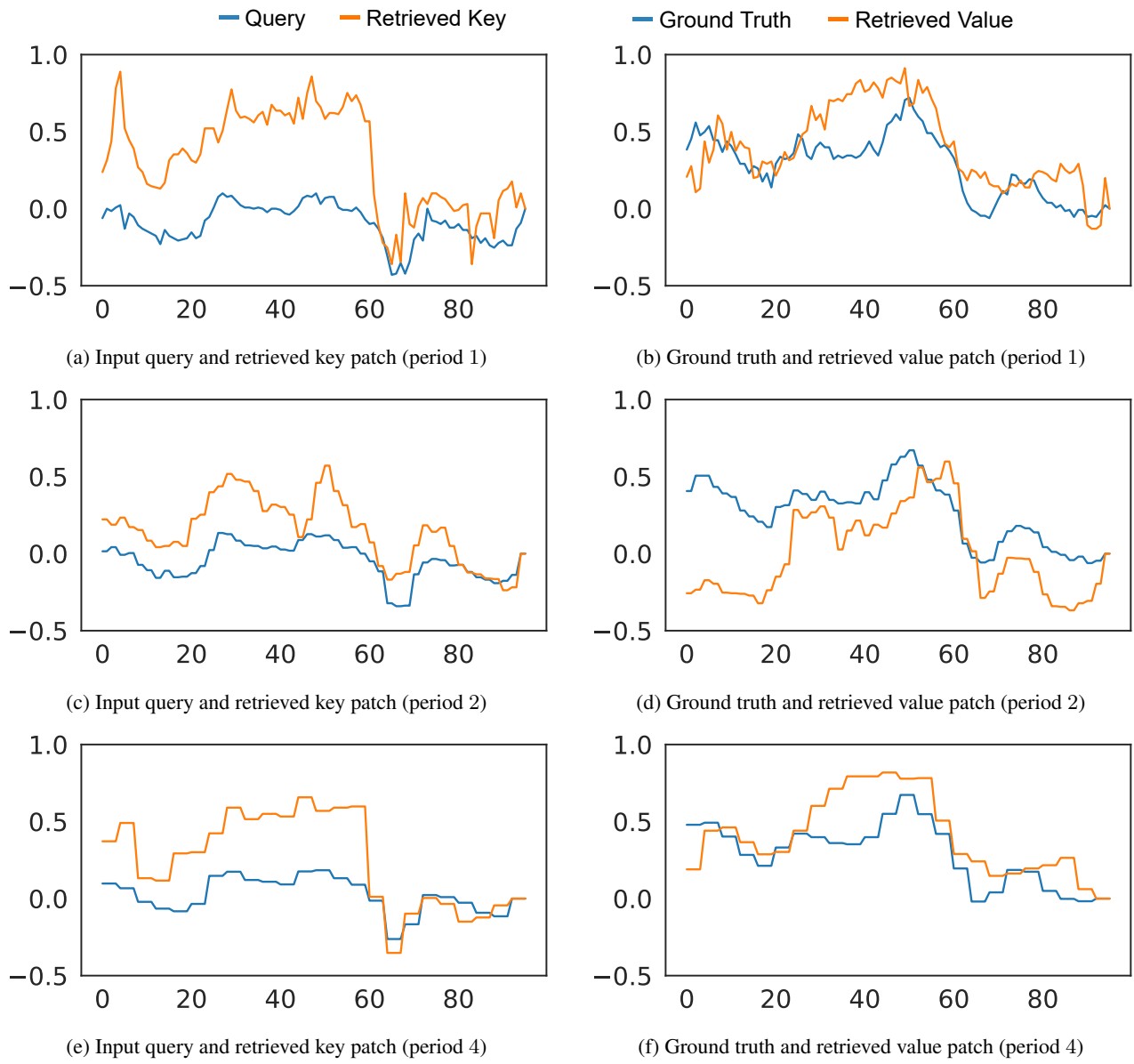

(a) Input query and retrieved key patch (period 1)

(b) Ground truth and retrieved value patch (period 1)

(c) Input query and retrieved key patch (period 2)

(d) Ground truth and retrieved value patch (period 2)

(e) Input query and retrieved key patch (period 4)

(f) Ground truth and retrieved value patch (period 4)

*Figure 7.* The example of our retrieval results on ETTh1 dataset. The key patches retrieved by period 1, 2, and 4 are compared with input query in (a), (c), and (e), respectively. The value patches retrieved by period 1, 2, and 4 are compared with ground truth in (b), (d), and (f), respectively. Note that the figures in the right side sequentially follows after the figures in the left side within the time series.

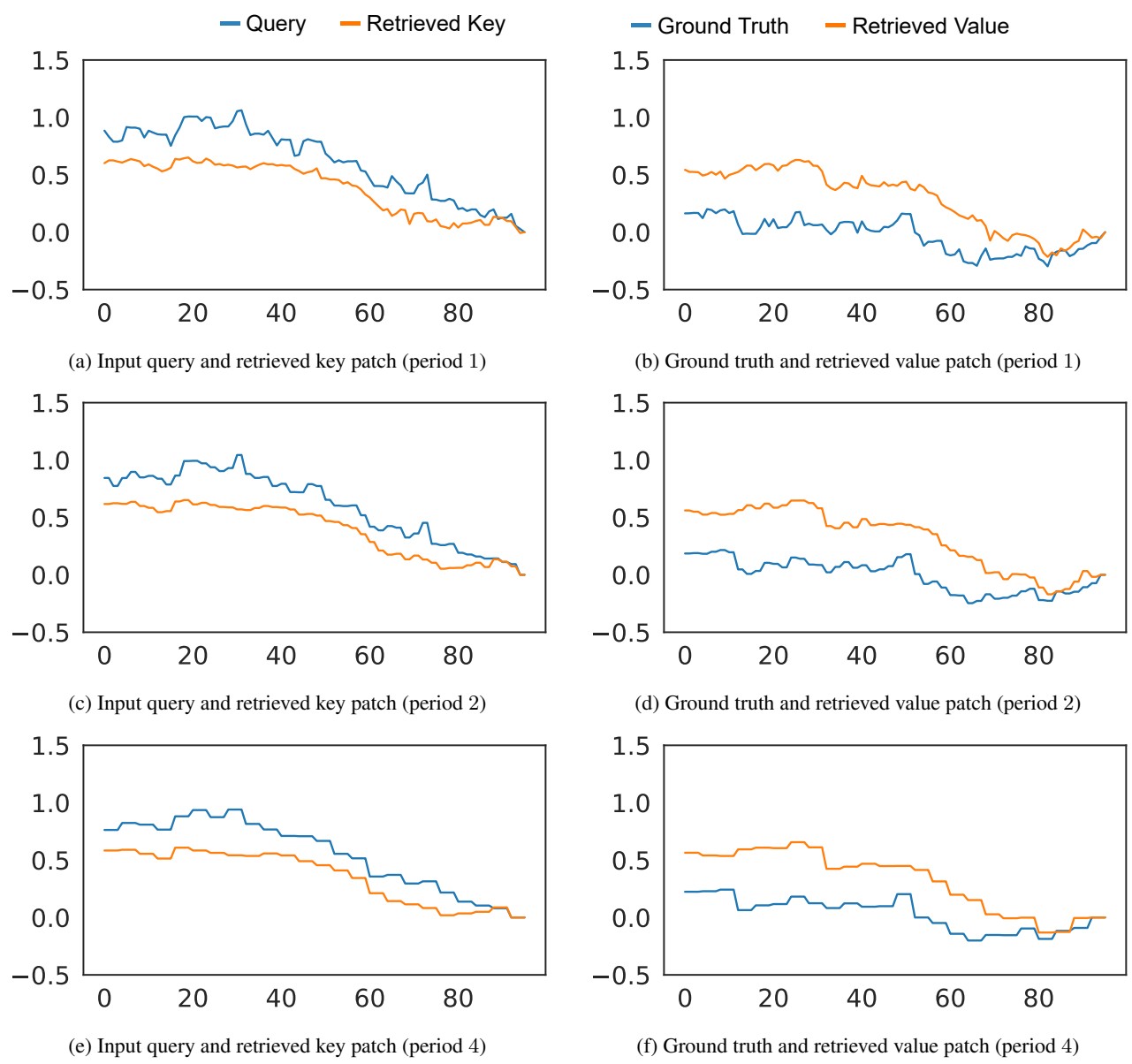

*Figure 8.* The example of our retrieval results on Exchange Rate dataset. The key patches retrieved by period 1, 2, and 4 are compared with input query in (a), (c), and (e), respectively. The value patches retrieved by period 1, 2, and 4 are compared with ground truth in (b), (d), and (f), respectively. Note that the figures in the right side sequentially follows after the figures in the left side within the time series.

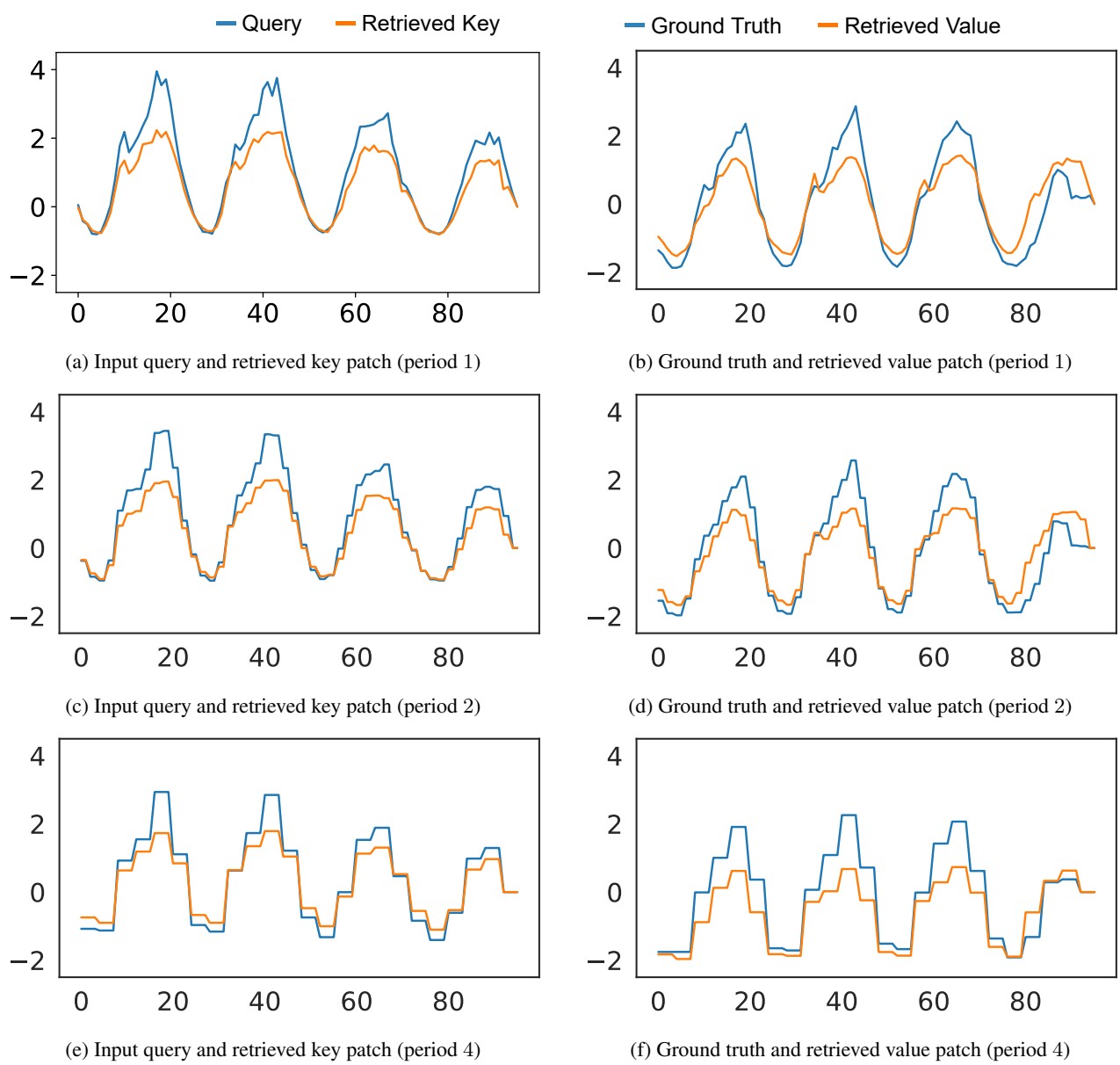

Figure 9. The example of our retrieval results on Traffic dataset. The key patches retrieved by period 1, 2, and 4 are compared with input query in (a), (c), and (e), respectively. The value patches retrieved by period 1, 2, and 4 are compared with ground truth in (b), (d), and (f), respectively. Note that the figures in the right side sequentially follows after the figures in the left side within the time series.

# G. Full Results

## G.1. Evaluation Results with MSE

*Table 16.* Full evaluation results with MSE are provided, with some baseline results excerpted from prior works (Wang et al., 2024; Nie et al., 2023).

| Methods | | Ours | TimeMixer | PatchTST | TimesNet | MICN | DLinear | FEDformer | Stationary | Autoformer | Informer |
|---|---|---|---|---|---|---|---|---|---|---|---|
| ETTh1 | 96 | 0.367 | 0.375 | 0.460 | 0.384 | 0.426 | 0.397 | 0.395 | 0.513 | 0.449 | 0.865 |
| | 192 | 0.411 | 0.429 | 0.512 | 0.436 | 0.454 | 0.446 | 0.469 | 0.534 | 0.500 | 1.008 |
| | 336 | 0.436 | 0.484 | 0.546 | 0.638 | 0.493 | 0.489 | 0.530 | 0.588 | 0.521 | 1.107 |
| | 720 | 0.467 | 0.498 | 0.544 | 0.521 | 0.526 | 0.513 | 0.598 | 0.643 | 0.514 | 1.181 |
| | Avg | 0.420 | 0.447 | 0.516 | 0.495 | 0.475 | 0.461 | 0.498 | 0.570 | 0.496 | 1.040 |
| ETTh2 | 96 | 0.276 | 0.289 | 0.308 | 0.340 | 0.372 | 0.340 | 0.358 | 0.476 | 0.346 | 3.755 |
| | 192 | 0.347 | 0.372 | 0.393 | 0.402 | 0.492 | 0.482 | 0.429 | 0.512 | 0.456 | 5.602 |
| | 336 | 0.376 | 0.386 | 0.427 | 0.452 | 0.607 | 0.591 | 0.496 | 0.552 | 0.482 | 4.721 |
| | 720 | 0.436 | 0.412 | 0.436 | 0.462 | 0.824 | 0.839 | 0.463 | 0.562 | 0.515 | 3.647 |
| | Avg | 0.359 | 0.365 | 0.391 | 0.414 | 0.574 | 0.563 | 0.437 | 0.526 | 0.450 | 4.431 |
| ETTm1 | 96 | 0.302 | 0.320 | 0.352 | 0.338 | 0.365 | 0.346 | 0.379 | 0.386 | 0.505 | 0.672 |
| | 192 | 0.329 | 0.361 | 0.390 | 0.374 | 0.403 | 0.382 | 0.426 | 0.459 | 0.553 | 0.795 |
| | 336 | 0.355 | 0.390 | 0.421 | 0.410 | 0.436 | 0.415 | 0.445 | 0.495 | 0.621 | 1.212 |
| | 720 | 0.406 | 0.454 | 0.462 | 0.478 | 0.489 | 0.473 | 0.543 | 0.585 | 0.671 | 1.166 |
| | Avg | 0.348 | 0.381 | 0.406 | 0.400 | 0.423 | 0.404 | 0.448 | 0.481 | 0.588 | 0.961 |
| ETTm2 | 96 | 0.164 | 0.175 | 0.183 | 0.187 | 0.197 | 0.193 | 0.203 | 0.192 | 0.255 | 0.365 |
| | 192 | 0.219 | 0.237 | 0.255 | 0.249 | 0.284 | 0.284 | 0.269 | 0.280 | 0.281 | 0.533 |
| | 336 | 0.275 | 0.298 | 0.309 | 0.321 | 0.381 | 0.382 | 0.325 | 0.334 | 0.339 | 1.363 |
| | 720 | 0.359 | 0.391 | 0.412 | 0.408 | 0.549 | 0.558 | 0.421 | 0.417 | 0.433 | 3.379 |
| | Avg | 0.254 | 0.275 | 0.290 | 0.291 | 0.353 | 0.354 | 0.305 | 0.306 | 0.327 | 1.410 |
| Electricity | 96 | 0.133 | 0.153 | 0.190 | 0.168 | 0.180 | 0.210 | 0.193 | 0.169 | 0.201 | 0.274 |
| | 192 | 0.149 | 0.166 | 0.199 | 0.184 | 0.189 | 0.210 | 0.201 | 0.182 | 0.222 | 0.296 |
| | 336 | 0.161 | 0.185 | 0.217 | 0.198 | 0.198 | 0.223 | 0.214 | 0.200 | 0.231 | 0.300 |
| | 720 | 0.197 | 0.225 | 0.258 | 0.220 | 0.217 | 0.258 | 0.246 | 0.222 | 0.254 | 0.373 |
| | Avg | 0.160 | 0.182 | 0.216 | 0.193 | 0.196 | 0.225 | 0.214 | 0.193 | 0.227 | 0.311 |
| Exchange | 96 | 0.091 | 0.095 | 0.084 | 0.107 | 0.102 | 0.081 | 0.148 | 0.111 | 0.197 | 0.847 |
| | 192 | 0.205 | 0.201 | 0.180 | 0.226 | 0.172 | 0.157 | 0.271 | 0.219 | 0.300 | 1.204 |
| | 336 | 0.353 | 0.350 | 0.510 | 0.367 | 0.272 | 0.305 | 0.460 | 0.421 | 0.509 | 1.672 |
| | 720 | 1.115 | 0.898 | 1.480 | 0.964 | 0.714 | 0.643 | 1.195 | 1.092 | 1.447 | 2.478 |
| | Avg | 0.441 | 0.386 | 0.564 | 0.416 | 0.315 | 0.297 | 0.519 | 0.461 | 0.613 | 1.550 |
| Illness | 24 | 2.076 | 1.896 | 1.319 | 2.317 | 2.684 | 2.215 | 3.228 | 2.294 | 3.483 | 5.764 |
| | 36 | 2.183 | 1.928 | 1.579 | 1.972 | 2.667 | 1.963 | 2.679 | 1.825 | 3.103 | 4.755 |
| | 48 | 2.073 | 2.132 | 1.553 | 2.238 | 2.558 | 2.130 | 2.622 | 2.010 | 2.669 | 4.763 |
| | 60 | 2.058 | 2.141 | 1.470 | 2.027 | 2.747 | 2.368 | 2.857 | 2.178 | 2.770 | 5.264 |
| | Avg | 2.097 | 2.024 | 1.480 | 2.139 | 2.664 | 2.169 | 2.847 | 2.077 | 3.006 | 5.137 |
| Solar | 96 | 0.192 | 0.189 | 0.265 | 0.373 | 0.257 | 0.290 | 0.286 | 0.321 | 0.456 | 0.287 |
| | 192 | 0.247 | 0.222 | 0.288 | 0.397 | 0.278 | 0.320 | 0.291 | 0.346 | 0.588 | 0.297 |
| | 336 | 0.240 | 0.231 | 0.301 | 0.420 | 0.298 | 0.353 | 0.354 | 0.357 | 0.595 | 0.367 |
| | 720 | 0.246 | 0.223 | 0.295 | 0.420 | 0.299 | 0.357 | 0.380 | 0.375 | 0.733 | 0.374 |
| | Avg | 0.231 | 0.216 | 0.287 | 0.403 | 0.283 | 0.330 | 0.328 | 0.350 | 0.593 | 0.331 |
| Traffic | 96 | 0.378 | 0.462 | 0.526 | 0.593 | 0.577 | 0.650 | 0.587 | 0.612 | 0.613 | 0.719 |
| | 192 | 0.391 | 0.473 | 0.522 | 0.617 | 0.589 | 0.598 | 0.604 | 0.613 | 0.616 | 0.696 |
| | 336 | 0.402 | 0.498 | 0.517 | 0.629 | 0.594 | 0.605 | 0.621 | 0.618 | 0.622 | 0.777 |
| | 720 | 0.434 | 0.506 | 0.552 | 0.640 | 0.613 | 0.645 | 0.626 | 0.653 | 0.660 | 0.864 |
| | Avg | 0.402 | 0.485 | 0.529 | 0.620 | 0.593 | 0.625 | 0.610 | 0.624 | 0.628 | 0.764 |
| Weather | 96 | 0.165 | 0.163 | 0.186 | 0.172 | 0.198 | 0.195 | 0.217 | 0.173 | 0.266 | 0.300 |
| | 192 | 0.211 | 0.208 | 0.234 | 0.219 | 0.239 | 0.237 | 0.276 | 0.245 | 0.307 | 0.598 |
| | 336 | 0.260 | 0.251 | 0.284 | 0.246 | 0.285 | 0.282 | 0.339 | 0.321 | 0.359 | 0.578 |
| | 720 | 0.327 | 0.339 | 0.356 | 0.365 | 0.351 | 0.345 | 0.403 | 0.414 | 0.419 | 1.059 |
| | Avg | 0.241 | 0.240 | 0.265 | 0.251 | 0.268 | 0.265 | 0.309 | 0.288 | 0.338 | 0.634 |

## G.2. Evaluation Results with MAE

*Table 17.* Full evaluation results with MAE are provided, with some baseline results excerpted from prior works (Wang et al., 2024; Nie et al., 2023).

| Methods | | Ours | TimeMixer | PatchTST | TimesNet | MICN | DLinear | FEDformer | Stationary | Autoformer | Informer |
|---|---|---|---|---|---|---|---|---|---|---|---|
| ETTh1 | 96 | 0.397 | 0.400 | 0.447 | 0.402 | 0.446 | 0.412 | 0.424 | 0.491 | 0.459 | 0.713 |
| | 192 | 0.427 | 0.421 | 0.477 | 0.429 | 0.464 | 0.441 | 0.470 | 0.504 | 0.482 | 0.792 |
| | 336 | 0.442 | 0.458 | 0.496 | 0.469 | 0.487 | 0.467 | 0.499 | 0.535 | 0.496 | 0.809 |
| | 720 | 0.478 | 0.482 | 0.517 | 0.500 | 0.526 | 0.510 | 0.544 | 0.616 | 0.512 | 0.865 |
| | Avg | 0.436 | 0.440 | 0.484 | 0.450 | 0.481 | 0.458 | 0.484 | 0.537 | 0.487 | 0.795 |
| ETTh2 | 96 | 0.344 | 0.341 | 0.355 | 0.374 | 0.424 | 0.394 | 0.397 | 0.458 | 0.388 | 1.525 |
| | 192 | 0.393 | 0.392 | 0.405 | 0.414 | 0.492 | 0.479 | 0.439 | 0.493 | 0.452 | 1.931 |
| | 336 | 0.425 | 0.414 | 0.436 | 0.452 | 0.555 | 0.541 | 0.487 | 0.551 | 0.486 | 1.835 |
| | 720 | 0.473 | 0.434 | 0.450 | 0.468 | 0.655 | 0.661 | 0.474 | 0.560 | 0.511 | 1.625 |
| | Avg | 0.409 | 0.395 | 0.412 | 0.427 | 0.532 | 0.519 | 0.449 | 0.516 | 0.459 | 1.729 |
| ETTm1 | 96 | 0.349 | 0.357 | 0.374 | 0.375 | 0.387 | 0.374 | 0.419 | 0.398 | 0.475 | 0.571 |
| | 192 | 0.367 | 0.381 | 0.393 | 0.387 | 0.408 | 0.391 | 0.441 | 0.444 | 0.496 | 0.669 |
| | 336 | 0.383 | 0.404 | 0.414 | 0.411 | 0.431 | 0.415 | 0.459 | 0.464 | 0.537 | 0.871 |
| | 720 | 0.413 | 0.441 | 0.449 | 0.450 | 0.462 | 0.451 | 0.490 | 0.516 | 0.561 | 0.823 |
| | Avg | 0.378 | 0.396 | 0.408 | 0.406 | 0.422 | 0.408 | 0.452 | 0.456 | 0.517 | 0.734 |
| ETTm2 | 96 | 0.256 | 0.258 | 0.270 | 0.267 | 0.296 | 0.293 | 0.287 | 0.274 | 0.339 | 0.453 |
| | 192 | 0.296 | 0.299 | 0.314 | 0.309 | 0.361 | 0.361 | 0.328 | 0.339 | 0.340 | 0.563 |
| | 336 | 0.336 | 0.340 | 0.347 | 0.351 | 0.429 | 0.429 | 0.366 | 0.361 | 0.372 | 0.887 |
| | 720 | 0.392 | 0.396 | 0.404 | 0.403 | 0.522 | 0.525 | 0.415 | 0.413 | 0.432 | 1.338 |
| | Avg | 0.320 | 0.323 | 0.334 | 0.333 | 0.402 | 0.402 | 0.349 | 0.347 | 0.371 | 0.810 |
| Electricity | 96 | 0.232 | 0.247 | 0.296 | 0.272 | 0.293 | 0.302 | 0.308 | 0.273 | 0.317 | 0.368 |
| | 192 | 0.247 | 0.256 | 0.304 | 0.322 | 0.302 | 0.305 | 0.315 | 0.286 | 0.334 | 0.386 |
| | 336 | 0.259 | 0.277 | 0.319 | 0.300 | 0.312 | 0.319 | 0.329 | 0.304 | 0.443 | 0.394 |
| | 720 | 0.297 | 0.310 | 0.352 | 0.320 | 0.330 | 0.350 | 0.355 | 0.321 | 0.361 | 0.439 |
| | Avg | 0.259 | 0.273 | 0.318 | 0.304 | 0.309 | 0.319 | 0.327 | 0.296 | 0.364 | 0.397 |
| Exchange | 96 | 0.209 | 0.214 | 0.203 | 0.234 | 0.235 | 0.203 | 0.278 | 0.237 | 0.323 | 0.752 |
| | 192 | 0.324 | 0.320 | 0.302 | 0.344 | 0.316 | 0.293 | 0.380 | 0.335 | 0.369 | 0.895 |
| | 336 | 0.431 | 0.427 | 0.531 | 0.448 | 0.407 | 0.414 | 0.500 | 0.476 | 0.524 | 1.036 |
| | 720 | 0.801 | 0.702 | 0.959 | 0.746 | 0.658 | 0.601 | 0.841 | 0.769 | 0.941 | 1.310 |
| | Avg | 0.441 | 0.416 | 0.499 | 0.443 | 0.404 | 0.378 | 0.500 | 0.454 | 0.539 | 0.998 |
| Illness | 24 | 0.956 | 0.860 | 0.754 | 0.934 | 1.112 | 1.081 | 1.260 | 0.945 | 1.287 | 1.677 |
| | 36 | 1.008 | 0.910 | 0.870 | 0.920 | 1.068 | 0.963 | 1.080 | 0.848 | 1.148 | 1.467 |
| | 48 | 0.972 | 0.956 | 0.815 | 0.940 | 1.052 | 1.024 | 1.078 | 0.900 | 1.085 | 1.469 |
| | 60 | 0.974 | 0.956 | 0.788 | 0.928 | 1.110 | 1.096 | 1.157 | 0.963 | 1.125 | 1.564 |
| | Avg | 0.977 | 0.920 | 0.807 | 0.931 | 1.086 | 1.041 | 1.144 | 0.914 | 1.161 | 1.544 |
| Solar | 96 | 0.251 | 0.259 | 0.323 | 0.358 | 0.325 | 0.378 | 0.341 | 0.380 | 0.446 | 0.323 |
| | 192 | 0.323 | 0.283 | 0.332 | 0.376 | 0.354 | 0.398 | 0.337 | 0.369 | 0.561 | 0.341 |
| | 336 | 0.300 | 0.292 | 0.339 | 0.380 | 0.375 | 0.415 | 0.416 | 0.387 | 0.588 | 0.429 |
| | 720 | 0.311 | 0.285 | 0.336 | 0.381 | 0.379 | 0.413 | 0.437 | 0.424 | 0.633 | 0.431 |
| | Avg | 0.296 | 0.280 | 0.333 | 0.374 | 0.358 | 0.401 | 0.383 | 0.390 | 0.557 | 0.381 |
| Traffic | 96 | 0.273 | 0.285 | 0.347 | 0.321 | 0.350 | 0.396 | 0.366 | 0.338 | 0.388 | 0.391 |
| | 192 | 0.277 | 0.296 | 0.332 | 0.336 | 0.356 | 0.370 | 0.373 | 0.340 | 0.382 | 0.379 |
| | 336 | 0.282 | 0.296 | 0.334 | 0.336 | 0.358 | 0.373 | 0.383 | 0.328 | 0.337 | 0.420 |
| | 720 | 0.297 | 0.313 | 0.352 | 0.350 | 0.361 | 0.394 | 0.382 | 0.355 | 0.408 | 0.472 |
| | Avg | 0.282 | 0.298 | 0.341 | 0.336 | 0.356 | 0.383 | 0.376 | 0.340 | 0.379 | 0.416 |
| Weather | 96 | 0.222 | 0.209 | 0.227 | 0.220 | 0.261 | 0.252 | 0.296 | 0.223 | 0.336 | 0.384 |
| | 192 | 0.264 | 0.250 | 0.265 | 0.261 | 0.299 | 0.295 | 0.336 | 0.285 | 0.367 | 0.544 |
| | 336 | 0.302 | 0.287 | 0.301 | 0.337 | 0.336 | 0.331 | 0.380 | 0.338 | 0.395 | 0.523 |
| | 720 | 0.355 | 0.341 | 0.349 | 0.359 | 0.388 | 0.382 | 0.428 | 0.410 | 0.428 | 0.741 |
| | Avg | 0.286 | 0.272 | 0.286 | 0.294 | 0.321 | 0.315 | 0.360 | 0.314 | 0.382 | 0.548 |

