# OpenReview forum: "Retrieval Augmented Time Series Forecasting"
_ICML.cc/2025/Conference — ICML 2025 poster_

### Official Review · Reviewer_43iV · 2025-03-10

**Overall Recommendation:** 2

**Summary:**

The author introduces retrieval-augmented methods into the time series forecasting problem, proposing Retrieval-Augmented Forecasting of Time-series (RAFT). During the forecasting process, the method retrieves the most similar historical windows from the training set to predict future data. The RAFT method has achieved favorable results on multiple datasets.

**Claims And Evidence:**

The article suggests that retrieval windows of different scales provide information at varying levels, such as local or global information. However, as mentioned in the article, when performing retrieval at different scales, RAFT scales the retrieval dataset, k, V, and input x, resulting in the sequences involved in the retrieval process corresponding to the same positions in the original sequence. It is unclear how this processing captures information at different scales. Furthermore, the paper does not include an ablation study to validate this claim, which may lead to confusion. I suggest that the authors conduct corresponding experiments to provide empirical evidence.

**Essential References Not Discussed:**

Not at all.

**Experimental Designs Or Analyses:**

1. The multiple periods used in RAFT, as mentioned in the "Claims and Evidence" section, lack experimental validation.
2. In Section 5.2,  the authors compare RAFT with and without the retrieval module to demonstrate the benefit of retrieval for rare events. Since retrieval augmentation inherently aims to improve forecasting accuracy, it is expected to enhance results. To better highlight the retrieval module’s ability to capture rare patterns, I suggest the authors conduct experiments on carefully synthesized datasets to compare RAFT against strong baselines.
3. Similarly, in Section 5.3, I recommend evaluating different baselines on the same dataset to substantiate the claim. Otherwise, comparing RAFT with and without retrieval alone may not be meaningful.
4. Section 5.4 states that retrieval is also effective for Transformer-based models and evaluates AutoFormer. Could the authors provide more details on how the retrieval module is incorporated into AutoFormer?

**Methods And Evaluation Criteria:**

The proposed method and evaluation criteria are appropriate for the time series forecasting problem.

**Other Comments Or Suggestions:**

1. Since setting the stride to 1 when obtaining K and V may lead to high computational complexity, could increasing the stride be beneficial? How would it impact the results?
2. During training, if the input sequence x is located in the middle of the training dataset S, does the retrieval module search for sequences both before and after x?

**Other Strengths And Weaknesses:**

Weakness see "Experimental Designs"

**Questions For Authors:**

See above comments.

**Relation To Broader Scientific Literature:**

There is a strong connection to retrieval augmentation in NLP.

**Theoretical Claims:**

The theoretical claims are basically correct.

---

> ### Author Rebuttal · Authors · 2025-04-01
>
> Thank you for your thoughtful comments. Please see our point-by-point response.
>
> > Claims and Evidence & W1. Rationale and ablation study of multiple periods in retrieval
>
> Thank you for raising this important point. First, we would like to clarify that our model performs retrieval across multiple periods (scales) and does not retrieve segments from identical positions across different periods. Instead, it retrieves segments at varying positions for each period based on their similarity to the input segment. Perhaps this misunderstanding arose because Figure 3 depicted keys and values from identical positions across multiple periods. In the revised manuscript, we will update Figure 3 to show different positions of retrieval in each period and revise the corresponding explanation on lines 193–197 of the right column.
>
> In addition to our response above, we conducted new ablation tests to empirically validate the effectiveness of multi-period retrieval by comparing performance with and without multiple periods. This new result is shown in the table below, which shows multi-period retrieval shows positive impacts for most datasets. We hope this answers your question and we will be happy to add these findings in the  revised manuscript in Appendix C.2.
>
> ||ETTh1|ETTh2|ETTm1|ETTm2|Electricity|Exchange Rate|Traffic|Weather|
> |-|:-:|:-:|:-:|:-:|:-:|:-:|:-:|:-:|
> |RAFT|0.367|0.276|0.302|0.164|0.133|0.091|0.378|0.165|
> |Without Retrieval|0.379|0.282|0.306|0.167|0.143|0.089|0.41|0.182|
> |Retrieval With 1 Period|0.369|0.276|0.303|0.164|0.133|0.088|0.379|0.168|
>
>
> > W2, W3. Baseline comparison in Section 5.2 and 5.3
>
> Thank you for the suggestion. As recommended, we have conducted additional experiments by comparing RAFT against strong baselines (represented in Table 1) on carefully synthesized datasets, extending our analyses presented in Sections 5.2 and 5.3. The comparison with the four strongest baselines selected based on winning ratio is summarized in the table below. Our results demonstrate that, in the experiments from Section 5.2, the proposed method showed comparable overall performance to the baselines but outperformed them in capturing rare patterns (e.g., occurrences = 1). In the Section 5.3 experiments, our method demonstrated superior performance in modeling less correlated patterns compared to all other baselines. We will include these new experimental results in the revised manuscript.
>
> |Pattern Occurrences (Section 5.2)|1|2|4|
> |--|:-:|:-:|:-:|
> |TimeMixer|0.236|0.217|0.228|
> |TimesNet|0.228|0.197|0.192|
> |MICN|0.228|0.233|0.203|
> |DLinear|0.264|0.255|0.250|
> |RAFT without Retrieval|0.259|0.231|0.234|
> |RAFT with Retrieval|0.221|0.206|0.213|
>
> | Pattern Occurrences (Section 5.3)|1|2|4|
> |-|:-:|:-:|:-:|
> |TimeMixer|0.286|0.230|0.225|
> |TimeNet|0.245|0.188|0.194|
> |MICN|0.254|0.245|0.245|
> |DLinear|0.318|0.206|0.280|
> |RAFT without Retrieval|0.269|0.265|0.189|
> |RAFT with Retrieval|0.185|0.182|0.159|
>
>
> > W4. Details of implementing RAFT in AutoFormer
>
> Rather than modifying the internal Transformer architecture to integrate our retrieval module, we directly appended retrieval results to AutoFormer’s predictions at the final stage. Here are some details: AutoFormer divides the input patches into trend and seasonal components, processes them separately, and combines them by addition at the final stage. Our retrieval results, $\sum_{p \in \mathcal{P}}{g^{(p)}(\tilde{\mathbf{v}}^{(p)})}$, are added at the final stage of AutoFormer. We will clarify these details in our revised manuscript.
>
> > Suggestion 1. Impact of stride on computational complexity and performance
>
> We agree that searching patches in extremely long time-series can be computationally intensive. To address this, we had introduced increasing a stride into the sliding window approach used during retrieval in Appendix D. Our experiments demonstrated that increasing the stride substantially decreases computational costs with minimal impact on performance. Results show that until stride 8, the performance loss is less than 5% while we could decrease the computational cost by 89%. As mentioned in our response to Reviewer 1 (W1), our method can be easily scalable to extremely long time series.
>
> > Suggestion 2. During training, if the input sequence x is located in the middle of the training dataset S, does the retrieval module search for sequences both before and after x?
>
> To prevent information leakage during training, any patches overlapping with the input sequence x were excluded from the retrieval candidates. After removing these overlapping patches, the retrieval module utilized patches from both before and after the sequence x. At inference time, we ensured a chronological split between the training and test sets, guaranteeing that the input sequence at test time always occurs after the entire training dataset. Therefore, there is no overlap, and the retrieval module leverages the full training set.

---

### Official Review · Reviewer_kZHz · 2025-03-11

**Overall Recommendation:** 4

**Summary:**

This paper proposes the RAFT framework, which leverages retrieval-augmented generation (RAG) to retrieve similar time series patterns and integrate them to enhance future predictions. The effectiveness of the RAFT framework has been evaluated on well-adopted time series benchmarks, with comparisons against state-of-the-art models.

**Claims And Evidence:**

Yes, the claims are well supported by clear and convincing evidence.

**Essential References Not Discussed:**

No.

**Experimental Designs Or Analyses:**

The design and validity of the experiment are well-founded.

**Methods And Evaluation Criteria:**

Yes, the evaluation criteria make sense for the problem.

**Other Comments Or Suggestions:**

No.

**Other Strengths And Weaknesses:**

The paper is well-written and clear, with a concise results comparison and convincing visualization of similar patterns. Well done!

**Questions For Authors:**

No.

**Relation To Broader Scientific Literature:**

This paper will have a broad impact on real-life applications of time series prediction models. It paves the way for applying retrieval-augmented generation (RAG) in the time series domain.

**Theoretical Claims:**

There is no theoretical claim in the paper.

---

> ### Author Rebuttal · Authors · 2025-04-01
>
> Thank you very much for acknowledging the novelty and wide applicability of our proposed retrieval method. We plan to further revise our manuscript by including additional experiments and clarifications in the revised manuscript.

---

### Official Review · Reviewer_AD5i · 2025-03-16

**Overall Recommendation:** 4

**Summary:**

This paper presents RAFT (Retrieval-Augmented Forecasting of Time Series), a method for enhancing time series forecasting models by retrieving relevant “patches” from the training dataset that match the current input pattern. These retrieved patches—subsequent future values corresponding to historically similar inputs—are then treated as an additional signal alongside the traditional model input. The authors demonstrate that RAFT boosts performance over numerous strong baselines (including Transformer-based approaches and simple linear or MLP models) across ten benchmark datasets. They also perform extensive analyses on synthetic data to highlight in which scenarios retrieval offers clear benefits (e.g., when patterns are rare or have low temporal correlation).

Main contributions

1. Proposing a retrieval-augmented architecture, specifically designed for single time-series forecasting, leveraging historical key-value “patches” in a memory-like module.
2. Showing consistent performance gains on real-world benchmarks (ETT, Electricity, Exchange, etc.) and describing scenarios where retrieval is particularly beneficial (e.g., repeated rare patterns).

**Claims And Evidence:**

* The authors claim that retrieval can help a model avoid “memorizing” rarely repeating patterns in its weights, instead leveraging a near-external memory approach to handle such patterns at inference time. Experimental results on synthetic data indeed show RAFT obtains substantial gains—particularly when short-term patterns reappear rarely and have random, less-correlated structures.
* They claim RAFT outperforms state-of-the-art baselines like Autoformer, FedFormer, PatchTST, etc. The tables (Table 1, for example) show RAFT surpassing these baselines on average MSE by a clear margin.

**Essential References Not Discussed:**

Overall, the references are comprehensive enough.

**Experimental Designs Or Analyses:**

1. The authors systematically test RAFT on 10 benchmarks commonly used in time-series forecasting. They vary the forecasting horizon and measure MSE/MAE. They also provide comparative results for 9+ baselines.
2. They do thorough ablations: e.g., removing the retrieval module, random retrieval, retrieval but no attention weighting, etc., to demonstrate each component’s impact.
3. Hyperparameter search is described (e.g., number of retrieved patches m, temperature for weighting, stride for scanning the dataset).
4. The authors present separate analyses on synthetic data to illustrate “rare patterns” and “less correlated random-walk patterns.” This is a strong demonstration of the retrieval approach’s value in difficult scenarios.

**Methods And Evaluation Criteria:**

Method Overview: RAFT uses a sliding window approach to generate “key-value” patches from the historical data, where keys are windows of size L and values are the subsequent F steps. Given a query window, RAFT calculates similarity (via Pearson correlation by default) to find top-m “key” matches, then aggregates the corresponding “value” patches. This aggregated retrieval is appended to the original input for final forecasting via a linear projection.

Evaluation: They compare MSE and MAE across multiple standard forecasting horizons (F = 96, 192, 336, 720) on widely used popular datasets. They provide ablations, sensitivity analyses, and comparisons to existing SOTA.

**Other Comments Or Suggestions:**

It would be interesting to see more real-world interpretability examples instead of the standard datasets.

**Other Strengths And Weaknesses:**

Strengths:

1. Strong empirical performance: RAFT  outperforms multiple strong baselines across standard real-world datasets.
2. Solid ablation and sensitivity studies: They carefully show how retrieval stride, temperature, offset normalization, etc. affect results, enhancing transparency.

Weakness:

1. Scalability: For extremely long series (millions of points), storing and searching patches might be expensive. More discussion on large-scale feasibility would be beneficial.
2. Confounding variables: Potential external or exogenous features (holidays, discrete events, etc.) might also be important to consider or retrieve.

**Questions For Authors:**

Q: As T grows large (say, 10 million+ data points), any efficient way of retrieval ?

Q: Does RAFT degrade gracefully when the time series is short (and thus historical patches are limited)? Are there diminishing returns?

Q: Could RAFT be extended to incorporate side-information retrieval from multiple correlated time series or external knowledge for each query patch?

**Relation To Broader Scientific Literature:**

1. Retrieval-based frameworks are used extensively in large language models to incorporate external knowledge. This work adapts that principle to time-series forecasting.
2. It aligns with the wave of “memory” or “nonparametric” approaches to handle patterns not frequently observed, reminiscent of k-NN or kernel-based time-series methods, but integrated into a deep learning pipeline.

**Theoretical Claims:**

This paper is applied.

---

> ### Author Rebuttal · Authors · 2025-04-01
>
> Thank you for your thoughtful comments. Please see our point-by-point response.
>
> > W1, Q1. Scalability to extremely long series.
>
> Thank you for highlighting this important point. We fully agree that searching patches in extremely long time-series can be computationally intensive. In light of this, we had introduced a stride into the sliding window approach used during retrieval to reduce the number of searched segments in the training data (see Appendix D). Our experiments demonstrated that increasing the stride substantially decreases computational costs with minimal impact on performance. For instance, using a stride of 8 reduces inference time by a factor of 8 (which could be even parallelized) with less than 5% degradation in performance.
>
> Another strategy could be to train a simple encoder to project the time segments of training data into low-dimensional dense embeddings and perform the search using cosine similarity (refer to Appendix C.1 - Cosine Similarity with Projection). According to our results presented in the Appendix C.1, this method maintains comparable performance with Pearson’s Correlation (default RAFT setting). This approach can significantly reduce inference-time complexity through precomputing dense embeddings of the training data segments. The entire process can be even parallelized, enabling it to handle datasets of a larger-scale such as billions of data points at inference. We can further accelerate the search efficiency via vector search solution through the approximated nearest neighbor (ANN) technique. We will incorporate this discussion into the revised manuscript.
>
> > W2, Q3. Extension to handling confounding variables (e.g., external or exogenous features)
>
> Thank you for this comment. Our model can be extended to incorporate external features or correlated time series by training encoders to project external information into low-dimensional dense embeddings within the same embedding space as the input features. One natural extension is to consider additional external time series. In this scenario, one encoder can project external segments into dense embeddings and another encoder can project input features into the same embedding space. When embeddings are aligned within a shared space, retrieval can use cosine similarity to identify relevant external segments for each input query patch. The retrieved external segments can then be used alongside the original input for enhanced prediction and encoders would be optimized jointly during training. This idea is consistent with our ablation study described in Appendix C.1 ("Cosine Similarity with Projection"). We will include this extension and its potential benefits in the revised manuscript.
>
> > Q2. Does RAFT degrade gracefully when the time series is short (and thus historical patches are limited)? Are there diminishing returns?
>
> It is correct that when the available time series is short, the number of historical patches for retrieval naturally decreases, potentially limiting performance gains. To investigate this phenomenon, we conducted additional experiments comparing our model with and without retrieval, varying the length of the training data from our benchmark datasets and evaluated performance on a fixed test set while adjusting the training data proportion to 20%, 60%, and 100%. Our results indicate that the impact of the training dataset's length on performance gains varies significantly across different datasets. However, our approach consistently outperformed the baseline model, even with limited historical data. We will include these findings and further discussion in our revised manuscript. We will include these findings and further discussion in our revised manuscript.
>
> | Dataset | Training data proportion | 20% | 60% | 100% |
> |:-:|-|:-:|:-:|:-:|
> | ETTh1 | RAFT without retrieval | 0.590 | 0.444 | 0.379 |
> |  | RAFT with retrieval | 0.562 | 0.428 | 0.367 |
> | ETTh2 | RAFT without retrieval | 0.251 | 0.255 | 0.282 |
> |  | RAFT with retrieval | 0.260 | 0.255 | 0.276 |
> | ETTm1 | RAFT without retrieval | 1.492 | 0.692 | 0.306 |
> |  | RAFT with retrieval | 0.975 | 0.684 | 0.302 |
> | ETTm2 | RAFT without retrieval | 1.364 | 0.608 | 0.167 |
> |  | RAFT with retrieval | 1.312 | 0.603 | 0.164 |
>
>
> > Suggestion. It would be interesting to see more real-world interpretability examples instead of the standard datasets.
>
> Thank you for the suggestion to showcase practical use cases with real-world data, which will enhance the interpretability of our model. We will include new interpretability examples using real-world scenarios, such as disease spread data from the COVID-19 pandemic (e.g., https://data.who.int/dashboards/covid19/) in the revised manuscript.

---

### Decision · Program_Chairs · 2025-05-01

**Decision:**

Accept (poster)

**Comment:**

This paper proposes RAFT, a retrieval-augmented method for time series forecasting, which enhances predictions by retrieving historical patterns similar to the current input and utilizing their subsequent values. The method demonstrates strong and consistent performance improvements over various state-of-the-art baselines across multiple benchmark datasets. Reviewers acknowledged the approach's novelty in applying retrieval to time series, its effectiveness, and the thoroughness of the empirical validation, including ablation and sensitivity studies. Initial concerns regarding scalability, the handling of multiple periods/scales, and specific experimental comparisons were effectively addressed by the authors during the rebuttal phase with additional experiments and clarifications. Therefore, I recommend acceptance of this paper.